# The discriminatory power of the T cell receptor

Johannes Pettmann[1,2†], Anna Huhn[1†], Enas Abu Shah[1,3†], Mikhail A Kutuzov[1], Daniel B Wilson[1,4], Michael L Dustin[3], Simon J Davis[2], P Anton van der Merwe[1], Omer Dushek[1]*

[1]Sir William Dunn School of Pathology, University of Oxford, Oxford, United Kingdom; [2]Radcliffe Department of Medicine, Medical Research Council Human Immunology Unit, Weatherall Institute of Molecular Medicine, University of Oxford, Oxford, United Kingdom; [3]Kennedy Institute of Rheumatology, University of Oxford, Oxford, United Kingdom; [4]Boston University, Department of Mathematics and Statistics, Boston, United States

**Abstract** T cells use their T cell receptors (TCRs) to discriminate between lower-affinity self and higher-affinity non-self peptides presented on major histocompatibility complex (pMHC) antigens. Although the discriminatory power of the TCR is widely believed to be near-perfect, technical difficulties have hampered efforts to precisely quantify it. Here, we describe a method for measuring very low TCR/pMHC affinities and use it to measure the discriminatory power of the TCR and the factors affecting it. We find that TCR discrimination, although enhanced compared with conventional cell-surface receptors, is imperfect: primary human T cells can respond to pMHC with affinities as low as $K_D \sim 1$ mM. The kinetic proofreading mechanism fit our data, providing the first estimates of both the time delay (2.8 s) and number of biochemical steps (2.67) that are consistent with the extraordinary sensitivity of antigen recognition. Our findings explain why self pMHC frequently induce autoimmune diseases and anti-tumour responses, and suggest ways to modify TCR discrimination.

*For correspondence:
omer.dushek@path.ox.ac.uk

†These authors contributed equally to this work

## Introduction

T cells use their T cell receptors (TCRs) to discriminate between lower-affinity self and higher-affinity non-self peptides presented on major histocompatibility complexes (pMHCs). This ability is the cornerstone of adaptive immunity and defects in this process can lead to autoimmunity. Although the strength of discrimination is widely believed to be near-perfect for the TCR (*Francois et al., 2013*; *Liu et al., 2014*; *Dushek and van der Merwe, 2014*; *Chakraborty and Weiss, 2014*; *Hong et al., 2018*; *Fernandes et al., 2019*; *Wu et al., 2019*; *Ganti et al., 2020*), systematic measurements to quantify it have not been performed.

Early influential studies using three murine TCRs suggested a sharp affinity threshold for T cell activation (*Hogquist et al., 1995*; *Alam et al., 1996*; *Alam et al., 1999*; *Kersh and Allen, 1996*; *Kersh et al., 1998b*; *Lyons et al., 1996*). Using T cells from the OT-I, 3.L2, and 2B4 transgenic TCR mice, it was shown that subtle changes to their cognate peptides, which apparently produced modest three- to fivefold decreases in affinity, abolished T cell responses even when increasing the peptide concentration by as much as 100,000-fold (*Hogquist et al., 1995*; *Alam et al., 1996*; *Alam et al., 1999*; *Kersh and Allen, 1996*; *Kersh et al., 1998b*; *Lyons et al., 1996*; *Altan-Bonnet and Germain, 2005*). Although this near-perfect discrimination based on affinity could be explained by a kinetic proofreading (KP) mechanism (*McKeithan, 1995*), it could not also account for the ability of T cells to respond to few pMHC ligands (high sensitivity; *Huang et al., 2013*; *Siller-Farfán and Dushek, 2018*).

Consequently, there has been a focus on identifying mechanisms that can simultaneously explain near-perfect discrimination and high sensitivity (*Altan-Bonnet and Germain, 2005*; *Francois et al., 2013*; *Liu et al., 2014*; *Dushek and van der Merwe, 2014*; *Chakraborty and Weiss, 2014*; *Hong et al., 2018*; *Fernandes et al., 2019*; *Wu et al., 2019*; *Ganti et al., 2020*). However, near-perfect discrimination is inconsistent with evidence that T cells can respond to lower-affinity self-antigens (*Yin et al., 2012*; *Bridgeman et al., 2012*), and moreover, that T cell-mediated autoimmunity is associated with increased expression of self-antigens (*Korem Kohanim et al., 2020*; *Wang et al., 2020*). There is thus a discrepancy between the current notion of near-perfect TCR discrimination and data on the role of T cell recognition of self-pMHC in human disease.

A key challenge in assessing discrimination is the accurate measurements of very weak TCR/pMHC affinities, with $K_D$ ranging from 1 to >100 µM (*van der Merwe and Davis, 2003*). A highly sensitive method for analysing molecular interactions is surface plasmon resonance (SPR), but even with this method, accurate measurements are difficult to make, especially at 37°C. In the case of OT-I, for example, measurements were performed at 37°C but high-affinity biphasic binding was observed (*Alam et al., 1999*), which has not been observed for other TCRs and may represent protein aggregates that often form at the high concentrations necessary for making these measurements. It follows that the reported small threefold change in affinity between the activating OVA and non-activating E1 ligands (*Alam et al., 1999*) may be a consequence of multivalent interactions. Indeed, more recent studies found the expected low-affinity monophasic binding for OT-I/OVA (*Stepanek et al., 2014*; *Liu et al., 2015*) and no detectable binding for OT-I/E1 (*Stepanek et al., 2014*). This raises the possibility that E1 does not activate T cells not because of near-perfect discrimination but simply because it does not bind the TCR. These studies highlight the challenges of accurately measuring TCR/pMHC affinities and underline their importance in our understanding of antigen discrimination.

Here, we introduce a new SPR protocol that can accurately determine ultra-low TCR/pMHC affinities at 37°C into the $K_D$ ~1 mM regime. We found that T cell responses were gradually lost as the affinity was decreased without a sharp affinity threshold and remarkably responses were detected to ultra-low-affinity pMHCs. By introducing a quantitative measure of discrimination, we are able to not only analyse our data but also analyse the published literature finding that the discriminatory power of the T cell receptor is imperfect yet remains above the baseline produced by other conventional surface receptors.

## Results
### Measurements of ultra-low TCR/pMHC affinities at 37°C

To assess discrimination, we first generated ligands to the anti-tumour 1G4 (*Chen et al., 2005*) and anti-viral A6 (*Garboczi et al., 1996*) TCRs recognising peptides on HLA-A*02:01. The standard SPR protocol is based on injecting the TCR at increasing concentrations over a pMHC-coated surface (*Figure 1A and B*) with the resulting steady-state binding response plotted over the TCR concentration (*Figure 1C*). This curve is fitted by a two-parameter Hill function to determine $B_{max}$ (the maximum response when all pMHCs are bound by TCR) and the $K_D$, which is the TCR concentration where binding is half the $B_{max}$. Therefore, an accurate determination of $K_D$ requires an accurate determination of $B_{max}$.

In the case of the 1G4 TCR binding to its cognate NY-ESO-1 peptide, this protocol produces $K_D \approx 7$ µM (*Figure 1A–C*, left column). However, the binding response curves do not saturate for lower-affinity pMHCs (*Figure 1A–C*, right column). Because of this, the fitted $B_{max}$ and therefore the fitted $K_D$ may not be accurate. Saturating the binding curves by increasing the TCR concentration is limited by the tendency of soluble recombinant proteins, including the TCR, to accumulate aggregates at high concentrations, which precludes accurate SPR measurements.

To determine $B_{max}$ when saturating pMHC with TCR was not feasible (i.e. for lower-affinity interactions), we generated a standard curve using the conformation-sensitive, pan-HLA-A/B/C antibody (W6/32) that only binds correctly folded pMHC (*Brodskys and Parham, 1982*). By injecting the W6/32 antibody at the end of each experiment (*Figure 1B*, black line), we were able to plot the fitted $B_{max}$ from higher-affinity interactions (where binding saturated) over the maximum W6/32 binding (*Figure 1D*). We observed a linear relationship even when including different TCRs binding different pMHC across multiple protein preparations immobilised at different levels. Together, this strongly suggested that

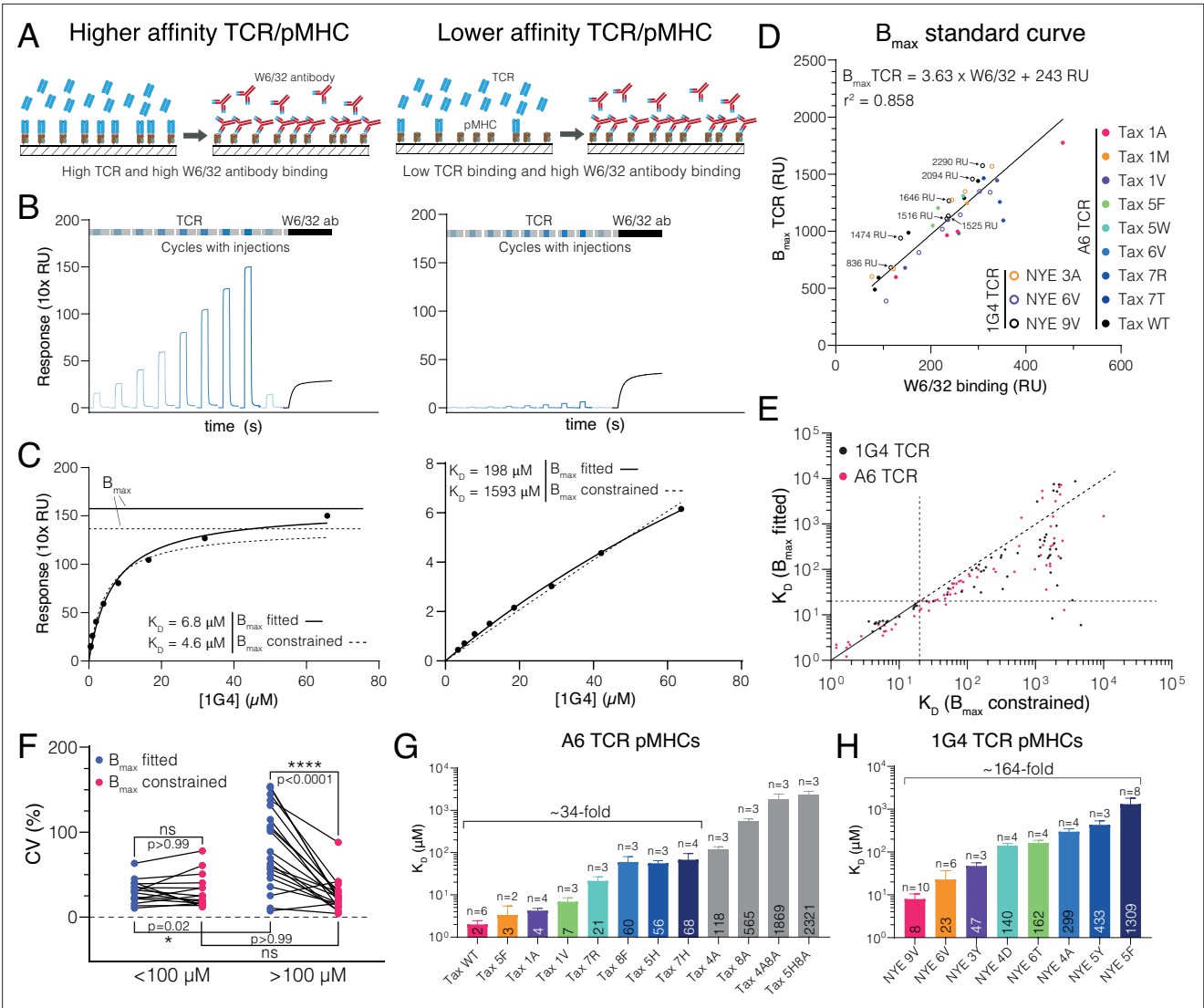

**Figure 1.** Measuring ultra-low T cell receptor (TCR)/peptides presented on major histocompatibility complex (pMHC) affinities using surface plasmon resonance (SPR) at 37°C using a constrained Bmax method. (**A–C**) Comparison of 1G4 TCR binding to a higher (left panels, NYE 9V) and lower (right panels, NYE 5F) affinity pMHC. (**A**) Schematic comparing TCR and W6/32 binding. (**B**) Example SPR sensograms showing injections of different TCR concentrations followed by the W6/32 antibody. (**C**) Steady-state binding response from (**B**) over the TCR concentration (filled circles) fitted to determine $K_D$ when Bmax is either fitted (standard method) or constrained (new method). Bmax obtained from either method is indicated for the high-affinity pMHC. For the low-affinity pMHC, the Bmax is out of the axes range (251 and 1671 RU for Bmax fitted and Bmax constrained, respectively). (**D**) Empirical standard curve relating W6/32 binding to fitted Bmax obtained using higher-affinity interactions. Immobilisation levels of NYE 9V are indicated showing that both W6/32 binding and fitted Bmax depend on the amount of pMHC immobilised. Although immobilisation levels are related to Bmax, they cannot be used directly because of variations in the fraction of inactive pMHC across different protein preparations (e.g. MHC that binds to the chip surface and hence contributes to immobilisation but is unfolded and cannot bind the TCR). Therefore, W6/32 binding provides an accurate proxy for the amount of active pMHC on the chip surface. (**E**) Correlation of $K_D$s obtained using the fitted and constrained methods. Each dot represents an individual measurement (n = 132; 61 for 1G4 TCR, 71 for A6 TCR). (**F**) Coefficient of variation for higher- (<100 μM) or lower-affinity (>100 μM) interactions. (**G**) Selected pMHC panel for A6 TCR. (**H**) Selected pMHC panel for 1G4 TCR. Mean values with SDs of $K_D$s are indicated in bars and ligands used for functional experiments in the main text are coloured. Data in (**A**) and (**B**) was double-referenced. The high- and low-affinity examples originate from different experiments.

The online version of this article includes the following figure supplement(s) for figure 1:

**Source data 1.** Double-referenced surface plasmon resonance data for *Figure 1*.

**Source data 2.** Fitted $K_D$s with the indicated method for the 1G4 TCR in SPR at 37°C.

**Source data 3.** Fitted $K_D$s with the indicated method for the A6 TCR in SPR at 37°C.

W6/32 and the TCR recognise the same correctly folded active pMHC population and justified the use of the standard curve to estimate $B_{max}$. While the level of W6/32 binding and $B_{max}$ is approximately proportional to the pMHC immobilisation level (see data for the NYE 9V pMHC in *Figure 1D*), the immobilisation level cannot be used to estimate $B_{max}$ because only a fraction of the pMHC immobilised is correctly folded and this fraction varies between protein preparations. We noted that W6/32 antibody binding was generally lower than TCR binding (e.g. *Figure 1B* and a slope of >1 in *Figure 1D*), which is unexpected because the molecular weight of the antibody is larger than the TCR. A likely explanation is that by injecting the antibody at a single concentration we have not saturated antibody binding. This is mitigated by ensuring that the same W6/32 antibody concentration is used and that $B_{max}$ is only interpolated within the standard curve.

We next fitted $K_D$ values for 132 interactions using the standard method where $B_{max}$ is fitted and the new method where $B_{max}$ is constrained to the value obtained using the standard curve (*Figure 1E*). In the new method, the only fitted parameter is $K_D$. Both methods produced similar $K_D$ values for higher affinities, validating the method (e.g. *Figure 1C*, left). In contrast, large (100-fold) discrepancies appeared for lower-affinity interactions, with the fitted $B_{max}$ method consistently underestimating the $K_D$. These large discrepancies were observed despite both methods providing a similar fit (e.g. *Figure 1C*, right). This suggested that for the fitted $B_{max}$ method different combinations of $B_{max}$ and $K_D$ can provide a fit of similar quality so that the fitted $K_D$ can exhibit large variations for the same interaction (also known as 'over-fitting'). We explored this by comparing the precision of both methods using the coefficient of variance (CV) of multiple measurements of the same TCR/pMHC combination. We found a similar CV for higher-affinity interactions (<100 μM $K_D$) and lower-affinity interactions when $B_{max}$ was constrained, but an increased CV for low-affinity interaction when $B_{max}$ was fitted (*Figure 1F*). Therefore, the standard method has lower precision for low-affinity interactions as a result of over-fitting.

We next used the new SPR method to accurately measure ultra-low affinities in order to identify panels of pMHCs that spanned the full physiological affinity range required to quantitate TCR discrimination (*Figure 1G and H*).

## Primary human T cells do not display a sharp affinity threshold and respond to ultra-low-affinity antigens

To quantify discrimination, we introduced the 1G4 TCR into quiescent naïve or memory CD8+ T cells and then co-cultured them with autologous monocyte-derived dendritic cells (moDCs) pulsed with each peptide (*Figure 2A*). Using surface CD69 as a marker for T cell activation, we found that lowering the affinity gradually reduced the response without the sharp affinity threshold suggested by near-perfect discrimination and, remarkably, responses were seen to ultra-low-affinity peptides, such as NYE 5F ($K_D$ = 1309 μM; see *Figure 2B and C*). To rule out preferential loading and/or stability of ultra-low-affinity peptides, we pulsed the TAP-deficient T2 cell lines with all peptides and found similar HLA upregulation, suggesting comparable loading and stability (*Figure 2—figure supplement 1*). We defined pMHC potency as the concentration of peptide required to reach 15% activation ($P_{15}$) in order to include lower-affinity pMHCs and found that it produced excellent correlations with $K_D$ (*Figure 2D and E*).

We observed similar results with T cell blasts (*Figure 2F and G*), which serve as an in vitro model for effector T cells and are commonly used in adoptive cell therapy. To independently corroborate discrimination with a second TCR, we used A6-expressing T cell blasts and again found a graded response (*Figure 2H*). However, potency for all pMHCs was lower and, therefore, responses were only observed for higher-affinity peptides with $K_D$ < 100 μM (*Figure 2H*, *Figure 2—figure supplement 2A and B*), which we attribute to the much lower expression of the A6 TCR (*Figure 2—figure supplement 2C and D*). Nonetheless, potency correlated with affinity (*Figure 2I and J*).

In order to quantify discrimination and sensitivity, we fitted the following power law to the data,

$$P_{15} = 10^C \times (K_D)^\alpha$$

where $C$ measures antigen sensitivity (y-intercept on the log-log plot) as the potency of a pMHC with $K_D$ = 1 μM (lower $C$ values indicate higher sensitivity), and α measures the discrimination power (slope on the log-log plot) as it quantifies the ability of a surface receptor to amplify changes in ligand affinity into potentially larger changes in ligand potency. Mechanistically, a receptor occupancy model,

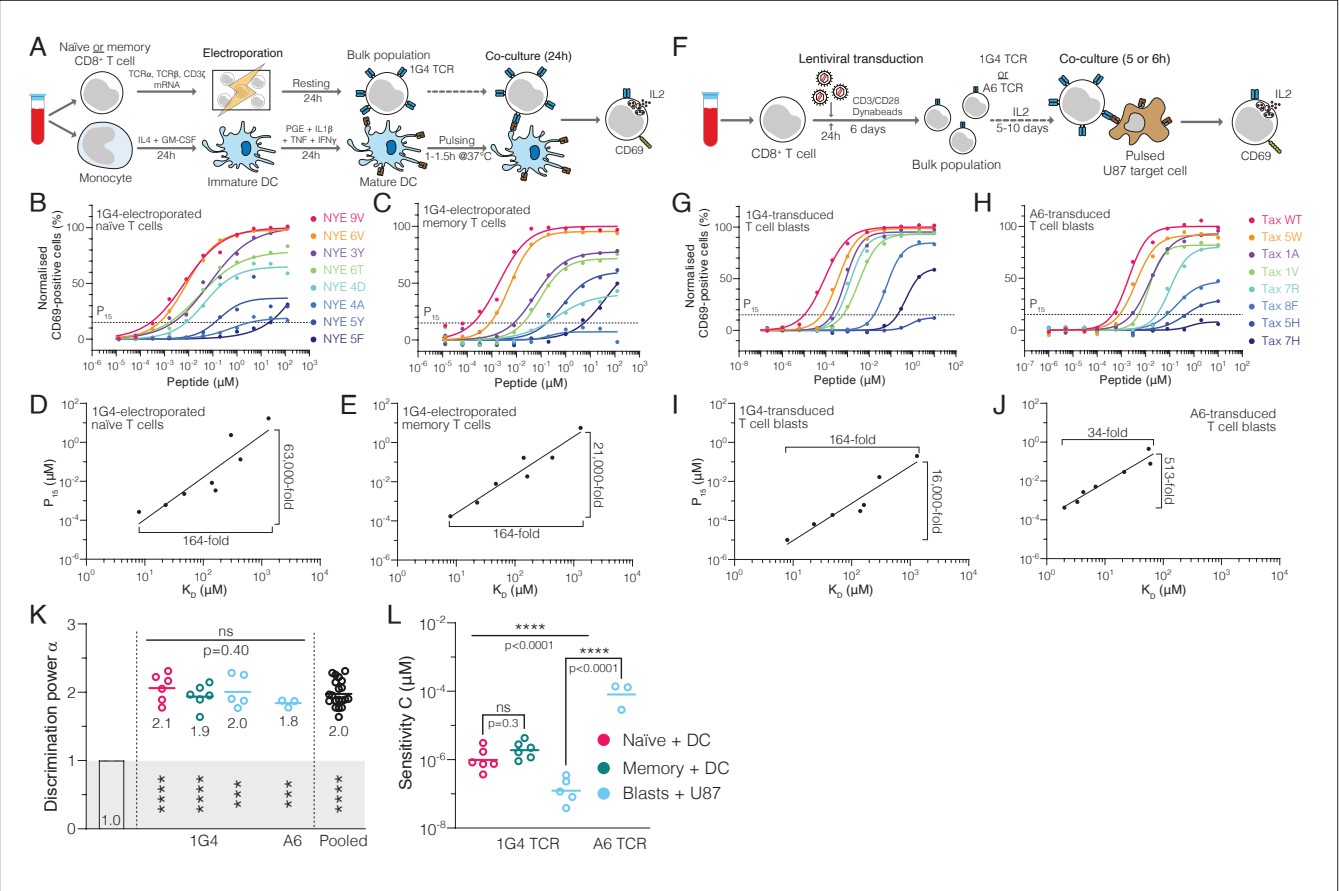

**Figure 2.** Naïve, memory, and blast human CD8+ T cells exhibit enhanced but imperfect discrimination. (**A**) Protocol for producing quiescent primary human naïve and memory CD8+ T cells interacting with autologous monocyte-derived dendritic cells as APCs. (**B, C**) Example dose-responses for naïve and memory T cells. Potency ($P_{15}$) is determined by the concentration of peptide eliciting 15% activation. (**D, E**) Examples of potency vs. $K_D$ fitted with a power law. Fold-change in $K_D$ and in potency derived from fits is shown. (**F**) Experimental protocol for producing primary human CD8+ T cell blasts interacting with the glioblastoma cell line U87 as APCs. (**G, H**) Example dose-responses and (**I, J**) potency vs. $K_D$ plots for T cell blasts expressing the indicated TCR. (**KL**) Comparison of the fitted discrimination power ($\alpha$) and fitted sensitivity ($C$). Shown are means with each dot representing an independent experiment (n = 3–6). (**K**) In grey the result of a statistical test vs. 1 is shown (p<0.0001 for naïve, memory and pooled, p=0.0002 for U87/1G4, p=0.0009 for U87/A6). 95% CI for pooled $\alpha$ in K is 1.9–2.1.

The online version of this article includes the following source data and figure supplement(s) for figure 2:

**Figure supplement 1.** All NYE peptides load similarly on T2 cells.

**Figure supplement 2.** T cells transduced with A6 T cell receptor (TCR) have low expression and do not respond to ultra-low-affinity peptides presented on major histocompatibility complexes (pMHCs).

**Figure supplement 3.** The discriminatory power based on cytokine production.

**Figure supplement 3—source data 1.** IL2 dose response data for *Figure 2—figure supplement 3*.

**Source data 1.** CD69 dose-response data for *Figure 2*.

where the response is proportional to the concentration of receptor/ligand complexes, produces $\alpha = 1$ (termed baseline discrimination as there is no amplification) whereas additional mechanisms are required to produce $\alpha > 1$ (termed enhanced discrimination). We observed enhanced discrimination powers (1.8–2.1) that were similar for naïve, memory, and blasted T cells and for both the 1G4 and A6 TCRs (*Figure 2K*), and when using IL-2 as a measure of T cell activation (*Figure 2—figure supplement 3A-C*). Despite these similar discrimination powers, we observed large ~1000-fold variation in antigen sensitivity (*Figure 2L*).

Taken together, while we found that the discriminatory power of the TCR was enhanced above baseline, we did not observe the previously reported sharp affinity threshold indicative of near-perfect discrimination.

## Systematic analysis reveals that the discriminatory power of the TCR is imperfect

Since α is a dimensionless measure of discrimination, we used it to compare the discriminatory power measured in this study with the apparently near-perfect discrimination suggested by earlier studies. We began by analysing the original three murine TCRs (*Figure 3A–C*). In the case of the OT-I TCR (*Figure 3A*), the T cell response was measured by target cell killing (*Hogquist et al., 1995*), and we defined potency as the peptide concentration producing 10% lysis ($P_{10}$) in order to include the E1 peptide variant. The original binding data was provided in a subsequent study (*Alam et al., 1999*). A plot of potency over $K_D$ revealed a very large discriminatory power ($\alpha = 10.5$), which reflects their finding that the E1 peptide variant had a $5 \times 10^6$-fold lower potency despite apparently having only a 3.5-fold lower affinity compared to the wild-type OVA peptide. We found similar large values of α (12, 18, and >5.1) for OT-I when using functional data from other studies (*Alam et al., 1996*; *Altan-Bonnet and Germain, 2005*; *Figure 3—source data 1* ID 1–4).

Similar to OT-I, the original data for the 3.L2 (*Kersh and Allen, 1996*; *Kersh et al., 1998b*) and 2B4 (*Lyons et al., 1996*) TCRs also produced large powers (*Figure 3B and C*). In the case of 3.L2, we plotted potency over $k_{off}$ instead of $K_D$ because $k_{on}$ was different between pMHCs (*Kersh et al., 1998b*; *Figure 3B*, bottom). Because of the small number of data points for these TCRs, the correlation plots used to determine α only reached statistical significance (p<0.05) for the 3.L2 TCR. Notwithstanding this limitation, this analysis supports the conclusions of these early mouse studies that TCR discrimination was near-perfect, with $\alpha \sim 9$ (see below).

The OT-I, 3.L2, and 2B4 transgenic mice continue to be instrumental in studies of T cell immunity, and as such, substantial data has been generated relating to these TCRs over the years, including new TCR/pMHC binding measurements. Revised SPR data for OT-I revealed no binding for the E1 peptide variant (*Stepanek et al., 2014*), and therefore, we could not use the original potency data. To produce an estimate of α for OT-I, we combined measurements of antigen potency (*Daniels et al., 2006*) and binding (*Stepanek et al., 2014*) that were now available for four peptides and found an appreciably lower discrimination power of 2.1 (*Figure 3D*). In the case of the 3.L2 TCR, revised SPR data for the original four peptide variants showed a wider variation in $K_D$ than originally reported (*Hong et al., 2015*). We re-plotted the original potency data over the revised $K_D$ value (as $k_{off}$ was not available for all peptides) and found a lower power of 3.2 (*Figure 3E*). Similarly, re-plotting the 2B4 TCR potency data over revised binding data (*Wu et al., 2002*) produced a lower discrimination power of 2.8 (*Figure 3F*). Although this calculation included only two data points, we identified two additional studies with 4–5 data points (*Birnbaum et al., 2014*; *Newell et al., 2011*) that also produced lower powers of 2.3 and 0.95 for 2B4 (*Figure 3—source data 1* ID 18 and 19).

Thus, estimates of discrimination powers of the OT-I, 3.L2, and 2B4 TCRs based on the early binding data were much higher (mean value of $\alpha \sim 9$) than those obtained when using more recent binding data (mean value of $\alpha = 2.2$) (*Figure 3I*), with the revised estimate being similar to the values obtained in this study for two TCRs (*Figure 2K*). This strongly suggests that discrepancies between the original mouse TCR data suggesting near-perfect discrimination ($\alpha \sim 9$) and our human TCR data suggesting imperfect discrimination ($\alpha = 2.0$) is a consequence of issues with the original SPR measurements.

Since many other mouse and human TCRs have been characterised over the past two decades, we used our approach to quantitate their discrimination powers. To be included in this study, a pMHC dose-response stimulation had to have been performed so that a measure of ligand potency could be determined and monomeric TCR/pMHC binding data ($K_D$ or $k_{off}$) also had to be available. We used studies that relied on different peptides that bound a single TCR, studies that relied on multiple TCRs that bound the same peptide, or studies that relied on a combination of both. We generated 51 potency plots (*Figure 3—figure supplement 2* and *Figure 3—figure supplement 3*) and extracted the discrimination power (*Figure 3—source data 1* ID 20–70). As representative examples, we show the mouse B3K506 TCR (*Figure 3G*) and the human 1E6 TCR (*Figure 3H*). Strikingly, analysis of these TCRs, and other mouse and human TCRs (*Figure 3J*), produced discrimination powers that were also significantly lower than those produced using the original mouse TCR data (*Figure 3I*). The variability across studies was not unexpected because they were not designed to accurately estimate α. Variability may be a result of the limited $K_D$ range and/or issues with estimating lower affinities. Nonetheless, combining all TCR data with the exception of the original mouse TCR data produced $\alpha = 2.0$ (95% CI of 1.5–2.4), in excellent agreement with our measurements. Therefore, a 5-fold decrease in

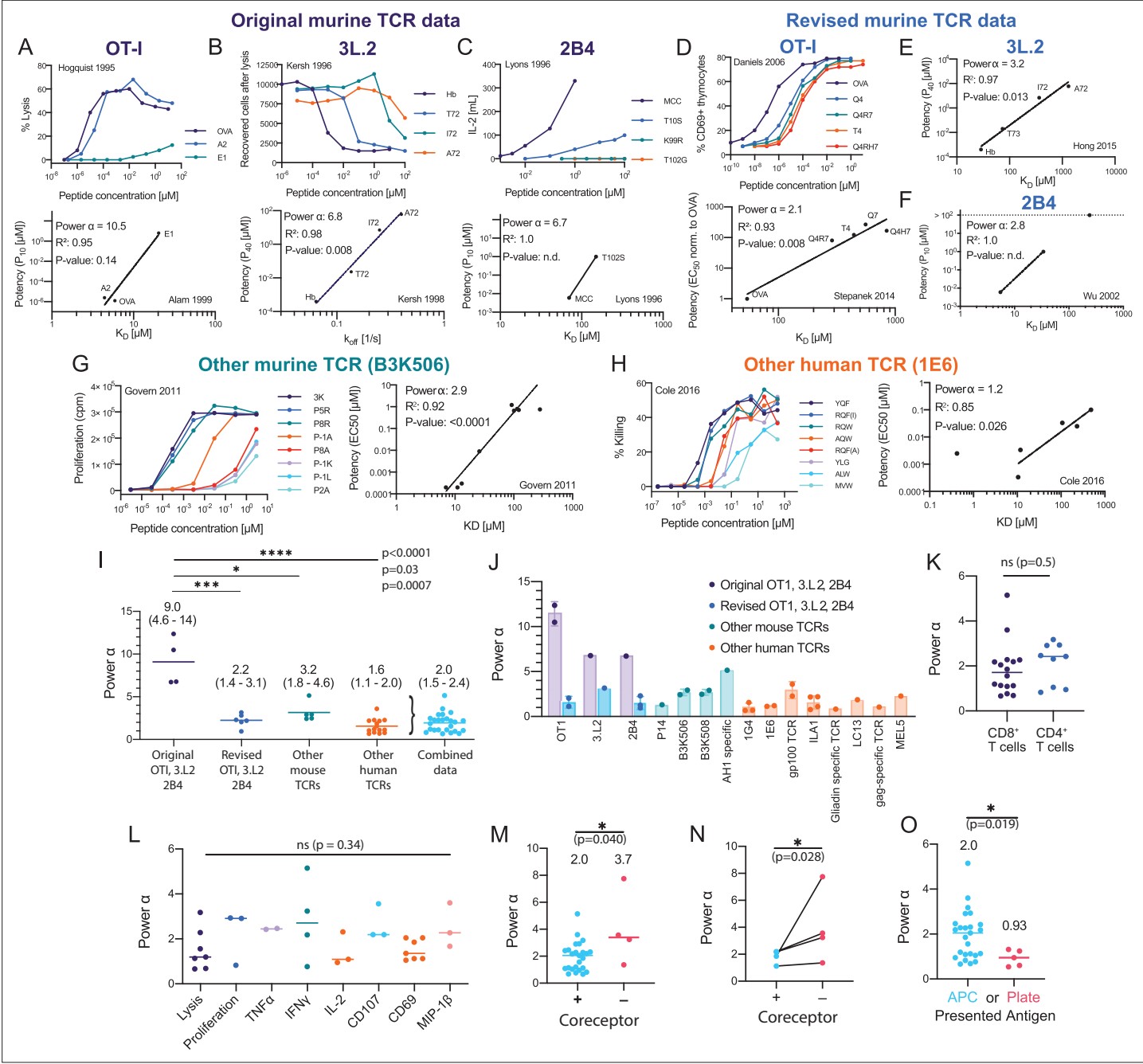

**Figure 3.** Systematic analyses show enhanced but imperfect discriminatory powers for the T cell receptor (TCR) that depend on the antigen-presenting surface. (**A–H**) T cell dose-responses and potency/affinity plots for (**A–C**) the original murine TCR data, revised analysis of the original murine TCRs using (**D**) new functional and binding data or (**E, F**) only new binding data, and examples of other (**G**) murine and (**H**) human TCRs. The highest affinity peptide ($K_D < 1$ μM) for the 1E6 TCR was excluded because it saturated the response and would have artificially lowered the fitted α (see Materials and methods for inclusion and exclusion criteria). IDs: 2 [**A**], 11 [**B**], 14 [**C**], 5 [**D**], 13 [**E**], 17 [**F**], 23 [**G**], and 42 [**F**]. (**I**) Comparison of discrimination powers with mean and 95% CI (combined data includes revised OTI, 3.L2, and 2B4 and other mouse and human data). (**J**) Discrimination powers shown in (**I**) parsed into each TCR. (**K**) Comparison between CD4[+] and CD8[+] T cells. (**L**) Comparison between different T cell responses. (**M**) Comparison between conditions with and without the CD4/CD8 co-receptors. (**N**) Comparison as in (**M**) but for paired data (where both conditions were present in the same study). (**O**) Comparison between the use of APCs or artificial plate surfaces to present antigens. Combined data is used in (**K, L**), (**M**) (+ co-receptor), and (**O**) (APC data).

The online version of this article includes the following source data and figure supplement(s) for figure 3:

**Source data 1.** Overview of discrimination powers for TCRs.

*Figure 3 continued on next page*

*Figure 3 continued*

**Figure supplement 1.** Potency over $K_D$ data for the original mouse T cell receptors (TCRs) (OT-I, 3.L2, and 2B4).

**Figure supplement 2.** Potency over $K_D$ data for other mouse T cell receptors (TCRs).

**Figure supplement 3.** Potency over $K_D$ data for other human T cell receptors (TCRs).

affinity can be compensated for by a 25-fold increase in antigen concentration for the TCR ($\alpha = 2$). While this is higher than the fivefold increase in concentration required by baseline discrimination ($\alpha = 1$), it is far lower than the unattainable 2-million-fold increase in concentration required by near-perfect discrimination ($\alpha = 9$). Taken together, this shows that the discriminatory power of the TCR is imperfect but enhanced above baseline.

## Factors affecting the discriminatory power of T cells

We next investigated factors that might affect the TCR discriminatory power. Using the literature data, we found no significant differences between CD4[+] or CD8[+] T cells (*Figure 3K*) or across different T cell responses (*Figure 3L*), which is consistent with a TCR proximal mechanism for discrimination. When we analysed studies where CD4/CD8 co-receptor binding was abolished (*Lo et al., 2019*; *Laugel et al., 2007*; *Burrows et al., 2010*), we found a significant increase in the discrimination power (*Figure 3M and N*), suggesting that the well-established role of co-receptors in increasing T cell sensitivity to antigen is accompanied by a decrease in discriminatory power.

We also identified studies where the antigen was presented on artificial surfaces in isolation (e.g. recombinant pMHC immobilised on plates; *Aleksic et al., 2010*; *Dushek et al., 2011*; *Lever et al., 2016*; *Abu-Shah et al., 2020*) and found that α decreased significantly from 2.0 on APCs to 0.93 on these surfaces (*Figure 3O*). Using our 1G4 T cell blasts, we confirmed that the discrimination power

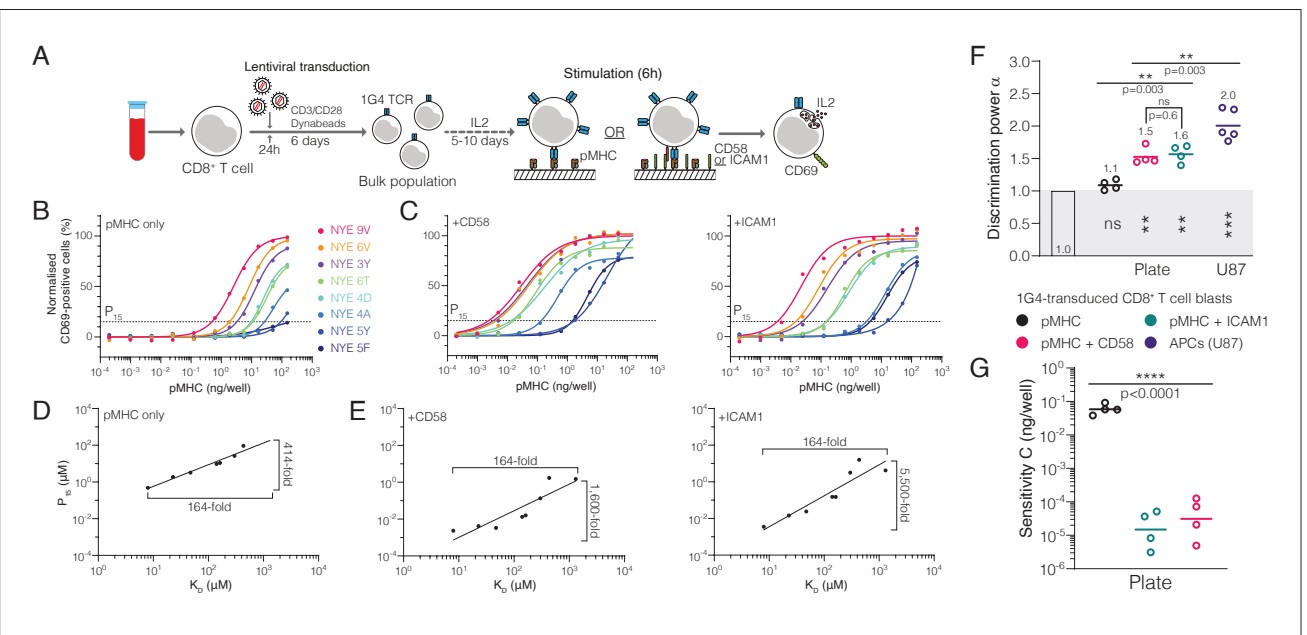

**Figure 4.** The T cell discriminatory power is enhanced by ligation of the receptors CD2 or LFA-1. (**A**) Protocol for stimulation of CD8[+] T cell blasts with plate-bound recombinant ligands. (**B, C**) Example dose-response curve for 1G4 T cell blasts stimulated with (**B**) peptides presented on major histocompatibility complex (pMHC) alone or (**C**) in combination with CD58 or ICAM1. (**D, E**) Potency derived from dose-response curves over $K_D$ showing the power function fit (**D**) with pMHC alone or (**E**) in combination with CD58 or ICAM1. (**F**) Comparison of the fitted discrimination power (α) and (**G**) fitted sensitivity (*C*). Shown are geometric means with each dot representing an independent experiment (n = 4–5). (**F**) In grey the result of a statistical test vs. 1 is shown (p=0.09 for pMHC, p=0.002 for CD58 and ICAM1, p=0.0002 for U87/1G4).

The online version of this article includes the following source data and figure supplement(s) for figure 4:

**Source data 1.** CD69 dose-response data for *Figure 4*.

**Figure supplement 1.** Engagement of CD2 or LFA-1 increases T cell receptor (TCR) downregulation.

**Figure supplement 1—source data 1.** Tetramer dose response data for *Figure 4—figure supplement 1*.

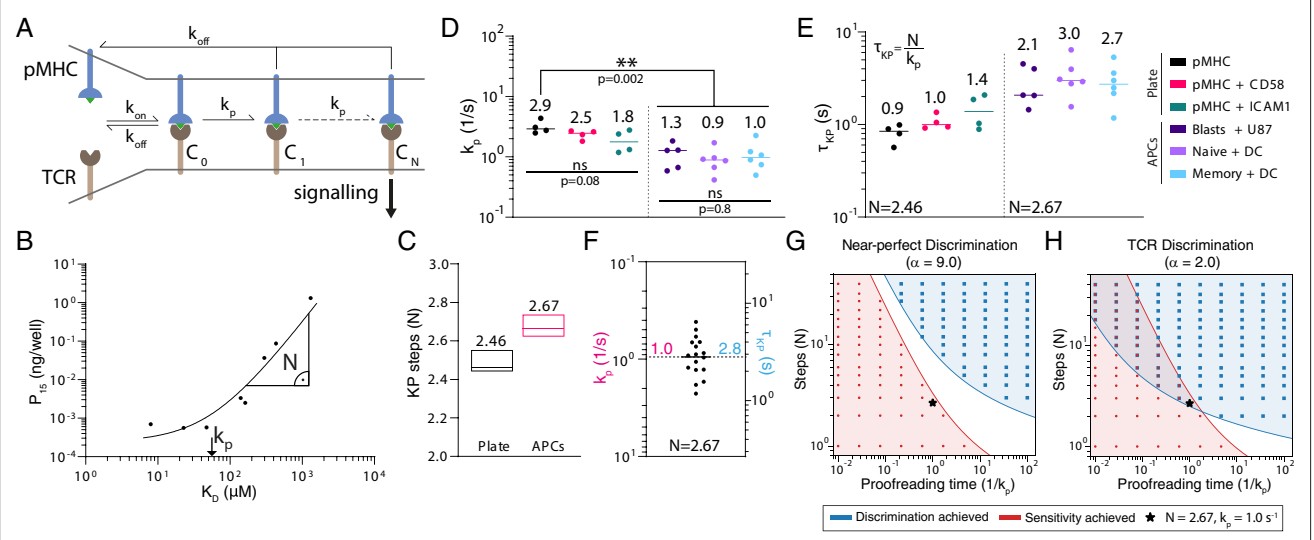

**Figure 5.** The kinetic proofreading mechanism explains T cell receptor (TCR) discrimination. (**A**) Schematic of the KP model. The KP time delay between initial binding (step 0) and signalling (step N) is $\tau_{KP} = N/k_p$. (**B**) Example fit of the KP model to data generated using CD8$^+$ blasts stimulated with peptides presented on major histocompatibility complex (pMHC) + ICAM1 showing that the fitted $k_p$ is near the $K_D$ threshold where potency saturates and $N$ is the slope away from this saturation point. (**C**) The fitted number of steps (median with min/max) was a global shared parameter for all plate or APC experiments. (**D**) The fitted KP rate was a local parameter for individual experiments. (**E**) The KP time delay calculated from N in (**C**) and individual $k_p$ values in (**D**). (**F**) Pooled 1G4 APC data are used to compute means of $k_p$ and $\tau_{KP}$ of 1.0 (95% CI: 0.7–1.2) and 2.8 (95% CI: 2.2–3.6), respectively. (**G,** **H**) Binary heatmaps showing when sensitivity (red) and discrimination (blue) are achieved for the indicated discrimination power. Results shown using stochastic simulations (dots) or deterministic calculations (continuous colours).

The online version of this article includes the following figure supplement(s) for figure 5:

**Figure supplement 1.** Direct fit of the kinetic proofreading model to potency data using the Approximate Bayesian Computation-Sequential Monte Carlo (ABC-SMC) method.

decreased from 2.0 when antigen was presented on APCs to 1.1 when presented as recombinant pMHC on plates (**Figure 4A,B,D,F**). This suggested that other factors, beyond TCR/pMHC, may be required for enhanced discrimination.

We hypothesised that co-signalling receptors CD2 and LFA-1 may be such factors because of their role in increasing ligand potency (**Bachmann et al., 1997**; **Bachmann et al., 1999**). Indeed, addition of recombinant ICAM1 (a ligand of LFA-1) or CD58 (the ligand to CD2) increased TCR downregulation (**Figure 4—figure supplement 1**) and antigen potency (**Figure 4C**) in this experimental system, consistent with previous reports using APCs (**Bachmann et al., 1997**; **Bachmann et al., 1999**). The potency plots highlighted that the 164-fold variation in $K_D$ was now amplified into a >1,600-fold variation in potency (**Figure 4E**) compared to only 414-fold when antigen was presented in isolation (**Figure 4D**). This is reflected in the discrimination power, which increased from 1.1 to >1.5 (**Figure 4F**). We noted that the 100-fold increase in antigen sensitivity is appreciably larger than previous reports (**Bachmann et al., 1997**; **Bachmann et al., 1999**) and likely reflects the reductionist system we have used where other co-signalling receptors cannot compensate (**Figure 4G**). These observations were reproduced using IL-2 as a measure of T cell activation (**Figure 2—figure supplement 3**). Therefore, engagement of the co-signalling receptors CD2 and LFA-1 enhances not only antigen sensitivity but also discrimination.

## The kinetic proofreading mechanism explains the discriminatory power of T cells

The KP mechanism proposes that a sequence of biochemical steps between initial pMHC binding (step 0) and TCR signalling (step N) introduces a proofreading time delay that tightly couples TCR signalling to the $k_{off}$ (or equivalently to $K_D$ if $k_{on}$ does not vary appreciably) of TCR/pMHC interactions (**Figure 5A**). Despite being introduced more than 20 years ago (**McKeithan, 1995**) and underlying all models of T cell activation (**Lever et al., 2014**), there are no estimates for two crucial parameters in

the model, namely the number of steps and the time delay for T cells discriminating antigens using APCs.

To determine the KP parameters, we fit the model simultaneously to all 1G4 potency data from the plate experiments (27 parameters fitted to 12 experiments with a total of 89 data points) or all 1G4 potency data from the APC experiments (37 parameters fitted to 17 experiments with a total of 126 data points). In both fits, we found excellent agreement (e.g. *Figure 5B*, *Figure 5—figure supplement 1A-B*) and, importantly, the fit method showed that $N$ and $k_p$ could be uniquely determined (*Figure 5—figure supplement 1C-H*). The value of $k_p$ was related to the $K_D$ value where potency saturated (i.e. showed no or modest changes as $K_D$ decreased) whereas the value of $N$ was the slope at much larger $K_D$ values (*Figure 5B*). Accurately determining both parameters required potency data spanning saturation to near-complete loss of responses, which can only be achieved by having a wide range of pMHC affinities down to very low affinities (high $K_D$). We found an unexpectedly small number of biochemical steps when fitting the APC data (2.67) and a similar value when independently fitting the plate data (*Figure 5C*). The fitted $k_p$ values were similar within the APC experiments but generally smaller than the plate experiments (*Figure 5D*), and because a similar number of steps was observed in both, this translated to the time delay which was longer on APCs (*Figure 5E*). Therefore, the higher discrimination power observed on APCs compared to the plate (*Figure 4F*) is a result of a longer time delay produced not by more steps but rather a slower rate for each step. This made conceptual sense because the number of steps is constrained by the signalling architecture whereas the rate of each step can be regulated. We combined the similar KP parameters for the APC data to provide an average time delay of $\tau_{KP} = 2.8$ s using $N = 2.67$ (*Figure 5F*).

Although the KP mechanism can explain our discrimination data, it has been previously argued that it cannot simultaneously explain the observed high sensitivity of the TCR for antigen (*Altan-Bonnet and Germain, 2005*; Dushek and *Dushek and van der Merwe, 2014*; *Chakraborty and Weiss, 2014*). We systematically varied the KP model parameters and determined whether discrimination and/or sensitivity were achieved for different levels of discrimination (*Figure 5G and H*). As in previous reports, we found that the KP mechanism could not simultaneously achieve sensitivity and near-perfect discrimination (*Figure 5G*). However, it readily achieved sensitivity and the revised imperfect discrimination that we now report, and interestingly, the 2.67 steps that we determined appear to be near the minimum number required to achieve this (*Figure 5H*). This may reflect the importance of maintaining a very short time delay so that antigen recognition can proceed rapidly allowing individual T cells to rapidly scan many APCs (*Altan-Bonnet and Germain, 2005*; Dushek and *Dushek and van der Merwe, 2014*; *Chakraborty and Weiss, 2014*).

## The discriminatory power of the TCR is higher than conventional surface receptors

Our finding that the discriminatory power of the TCR is only modestly enhanced above baseline raises the important question of whether it is unique in its ligand discrimination abilities. To answer this question, we identified studies that allowed us to estimate the discrimination power for cytokine receptors, receptor-tyrosine-kinases (RTKs), G-protein coupled receptors (GPCRs), chimeric antigen receptors (CARs), and B cell receptors (BCRs) (*Figure 6A–E*). Out of 30 calculations, we found 21 significant correlations between potency and $K_D$ (or $k_{off}$) that allowed us to estimate α (*Figure 6—source data 1*). We found that the discrimination powers of cytokine receptors, RTKs, GPCRs, and CARs were all at or below 1, and as a group, their discrimination powers were significantly lower than the TCR (*Figure 6F*). We identified only a single study for the BCR that could be used to compute α and report a preliminary discrimination power of 1.3, which is intermediate between the TCR and other receptors. Therefore, the TCR appears to be unique in its enhanced ligand discriminatory powers.

## Discussion

In contrast to the prevailing view that the TCR exhibits near-perfect discrimination, we have shown here that the discriminatory power of the TCR is imperfect and that it is able to respond to ultra-low-affinity antigens. Our estimates of TCR discrimination were facilitated by the development of a revised SPR method to accurately measure TCR/pMHC affinities.

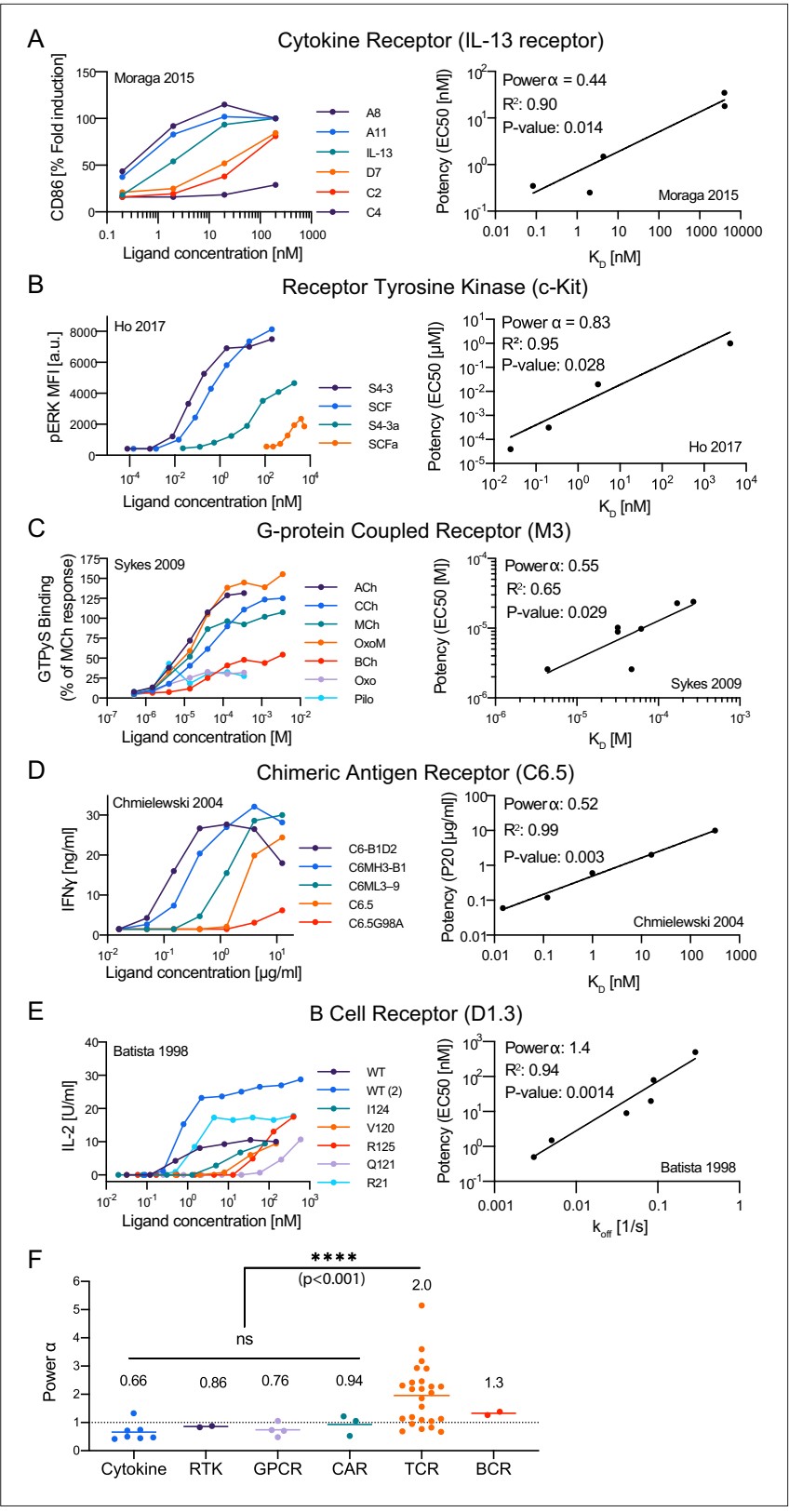

**Figure 6.** The discriminatory power of the T cell receptor (TCR) is higher than conventional surface receptors. (**A–E**) Representative dose-response (left column) and potency over KD or $k_{off}$ (right column) for the indicated surface receptor. IDs: 5 [**A**], 15 [**B**], 20 [**C**], 25 [**D**], 29 [**E**]. (**F**) Discrimination powers for the indicated receptor. Data for the TCR as in *Figure 3I* (combined data).

*Figure 6 continued on next page*

*Figure 6 continued*

The online version of this article includes the following source data and figure supplement(s) for figure 6:

**Source data 1.** Overview of discrimination powers for other (non-TCR) surface receptors.

**Figure supplement 1.** Potency over KD data for other (non-T cell receptor) receptors.

---

The KP mechanism was able to explain both the high antigen sensitivity and the discrimination power of the TCR. This was achieved by a few steps (2.67) and a short proofreading time delay (2.8 s). This time delay is at the shorter end of the value estimated using pMHC tetramers (8 s with 95% CI: 3–19 s) (*Yousefi et al., 2019*) and consistent with the 4 s time delay between pMHC binding and LAT phosphorylation (*Huse et al., 2007*). The small number of steps is reasonable because, although the TCR complex undergoes a large number of biochemical modifications (*Chakraborty and Weiss, 2014*), only those that must be sequential contribute. It follows that multiple ITAMs acting in parallel would not extend the proofreading chain. In support of this, the number of steps we estimate here for the TCR with 10 ITAMs is the same as the number recently reported for a CAR with 6 ITAMs ($2.7 \pm 0.5$) (*Tischer and Weiner, 2019*).

The finding that the number of KP steps is fractional (2.67) may suggest that at least one intermediate proofreading step is not instantly reversible. For example, a proofreading chain with three steps where the first step can be sustained after ligand unbinding would generate a population of TCRs that required only two steps before productive signalling. Depending on the relative concentration of this TCR population, the apparent number of steps can be between 3 and 2. Therefore, the fractional number of steps that we have observed suggests that one (or more) KP step may be sustained upon pMHC unbinding, which may represent the time delay between pMHC unbinding and the dephosphorylation of the TCR signalling complex and/or the unbinding of ZAP70 (*Wang et al., 2010*; *Goyette et al., 2020*).

Our finding that the discriminatory power of the TCR is enhanced compared with conventional receptors raises the question as to the underlying mechanism. One distinct feature of the TCR is that recognition occurs at a cell-cell interface and is assisted by co-signalling receptors such as CD2 and LFA-1, which appear to be required for enhanced discrimination. Our preliminary observation that the BCR may also exhibit enhanced discrimination suggests a role for ITAM-based signalling in enhanced discrimination. While our finding that ITAM-based CARs did not exhibit enhanced discrimination argues against this, CARs are artificial chimeric molecules with defects in ITAM signalling (*Gudipati et al., 2020*).

Although ligand potency usually correlates with solution or three-dimensional (3D) affinity measured by SPR, there are occasional exceptions. In one example, a structural explanation was provided for a pMHC that could bind the TCR but could not activate T cells; it exhibited an unusual docking geometry that prevented co-receptor binding (*Adams et al., 2011*). In another example, it was suggested that mechanical forces could affect the TCR binding affinity to different ligands in a different way (*Liu et al., 2014*; *Sibener et al., 2018*). Finally, in a third example, it was shown that the surface or two-dimensional (2D) TCR/pMHC binding parameters measured within the T cell contact interface predicted the T cell response more accurately compared to the 3D binding parameters measured in SPR (*Huang et al., 2010*). However, this was based on the earlier inaccurate SPR data for the OT-I system, which was the only data available at the time. A subsequent study found that the 2D and 3D binding parameters for the 1E6 TCR were equally accurate at predicting the T cell response (*Cole et al., 2016*). Taken together, these studies suggest that there are likely to be occasional exceptions where 3D binding properties do not correlate with potency. This may partly explain the lack of correlation between potency and 3D affinity reported in a subset of the published studies we have analysed (*Figure 3—source data 1*).

We found that the basic KP mechanism was sufficient to accurately capture antigen discrimination within the physiological affinity range and when antigens are presented in the context of self pMHCs on autologous APCs. However, it is known that the basic KP mechanism alone cannot explain the phenomena of antagonism or optimal affinity. Antagonism is a phenomena where lower-affinity pMHCs, which do not induce T cell responses on their own, are able to inhibit T cell activation by agonist pMHCs (*Ma et al., 1999*; *Yang and Grey, 2003*; *Altan-Bonnet and Germain, 2005*; *Stone et al., 2011*). This can be explained by augmenting the KP mechanism with feedbacks (*Altan-Bonnet*

and Germain, 2005; Francois et al., 2013; Lever et al., 2014). In studies that used supra-physiological TCR/pMHC affinities, it was observed that T cell responses eventually decreased as the affinity increased (Kalergis et al., 2001; Corse et al., 2010; Irving et al., 2012; Lever et al., 2016). This optimal pMHC affinity can be explained by augmenting the KP mechanism with limited signalling (Lever et al., 2014). In the future, including data using supra-physiological and/or antagonist antigens can be used to calibrate a KP model augmented with limited signalling and/or feedbacks.

To study discrimination, we have introduced the discriminatory power (α) because it can quantify discrimination, independently from antigen sensitivity, from experimental studies. Previously, the term specificity has been used to refer to this discriminatory concept (Altan-Bonnet and Germain, 2005; Francois et al., 2013; Dushek and Dushek and van der Merwe, 2014; Ganti et al., 2020). However, specificity is also commonly used to mean the opposite of promiscuity (i.e. the ability of T cells to respond to many different peptides). To avoid ambiguity, we suggest that specificity and promiscuity are used to refer to the tolerance of peptide sequence diversity while discrimination is used to refer to the tolerance of changes in TCR/pMHC binding parameters. Using this terminology, our analysis suggests that co-receptors decrease the discriminatory power of the TCR (Figure 3M and N) and published data has demonstrated that co-receptors can increase the promiscuity of the TCR (Wooldridge et al., 2010).

The imperfect discriminatory power of the TCR has important functional consequences. Under the assumption of near-perfect TCR discrimination, T cell-mediated autoimmunity is often viewed as a defect in thymic-negative selection and/or peripheral tolerance mechanisms (Yin et al., 2012). However, with an imperfect discriminatory power of $\alpha = 2$, the 10–100-fold lower affinity reported for autoreactive TCRs binding their self antigens (Yin et al., 2012; Bridgeman et al., 2012) means that they can become activated if their self antigens increase in expression by 100–10,000-fold. This suggests that T cell autoimmunity can arise by inappropriate increases in expression of self antigens, and such increases have recently been implicated in T cell-mediated autoimmunity (Korem Kohanim et al., 2020; Wang et al., 2020). T cells also have important roles in eliminating tumour cells but their therapeutic use is often limited by toxicities to lower-affinity off-tumour antigens (e.g. Cameron et al., 2013). The factors we have identified that control antigen discrimination, together with the proposed mechanisms that can generate near-perfect discrimination (Chan et al., 2003; Altan-Bonnet and Germain, 2005; Francois et al., 2013; Dushek and van der Merwe, 2014; Ganti et al., 2020), may enable the engineering of T cells with improved discriminatory powers that selectively reduce responses to lower-affinity off-tumour antigens.

## Materials and methods

**Key resources table Key resources table**

| Reagent type (species) or resource | Designation | Source or reference | Identifiers | Additional information |
|---|---|---|---|---|
| Cell line (human) | U87 | Vincenzo Cerundolo | | University of Oxford, UK |
| Cell line (human) | Freestyle 293 F | Thermo Fisher Scientific | RRID:CVCL_D603 | For protein production by transient transfection. |
| Cell line (human) | Lenti-X 293T | Takara Bio | RRID:CVCL_4401 | For production of lentivirus. |
| Transfected construct (human) | pTT3-ecdCD58 | This paper | | Plasmid for production of recombinant, soluble CD58 through transient transfection of mammalian cells. |
| Transfected construct (human) | pTT3-ecdICAM1 | This paper | | Plasmid for production of recombinant, soluble ICAM1 through transient transfection of mammalian cells. |
| Transfected construct (human) | pTT3-BirA-FLAG | Addgene | RRID:Addgene_64395 | Plasmid for in-flask biotinylation by co-transfection. Bushell et al., 2008. |
| Antibody | Anti-human CD69 (mouse monoclonal) | Biolegend | RRID:AB_314839; RRID:AB_528869; RRID:AB_2561909; RRID:AB_528871 | Colours: FITC, AF488, BV421, AF647; dilution: (1:200); clone: FN50. |

*Continued on next page*

*Continued*

| Reagent type (species) or resource | Designation | Source or reference | Identifiers | Additional information |
|---|---|---|---|---|
| Antibody | Anti-human CD45 (mouse monoclonal) | Biolegend | RRID:AB_2561357; RRID:AB_2563466 | Colours: BV421, BV711; dilution: (1:200); clone: HI30. |
| Antibody | Anti-HLA-A2 (mouse monoclonal) | Biolegend | RRID:AB_2721523; RRID:AB_1877227 | Colours: BV421, PE; dilution: (1:100–1:200); clone: BB7.2. |
| Antibody | Human TruStain Fc block | Biolegend | RRID:AB_2818986 | Dilution: (1:100). |
| Antibody | W6/32 | Biolegend | RRID:AB_314871 | Unconjugated; for SPR; lot: B233942. |
| Recombinant DNA reagent | pLEX-A6 | This paper | | Lentiviral transfer plasmid based on pLEX307. See *Supplementary file 1* for insert sequence. |
| Recombinant DNA reagent | pHR-1G4 | This paper | | Lentiviral transfer plasmid with EF1α promoter for transduction of the 1G4 TCR into T cells. See *Supplementary file 1* for insert sequence. |
| Recombinant DNA reagent | A6α | This paper | | Soluble A6 alpha chain for production in *Escherichia coli* and in vitro refolding. See *Supplementary file 2* for insert sequence. |
| Recombinant DNA reagent | A6β-His | This paper | | Soluble A6 beta chain for production in *E. coli* and in vitro refolding. See *Supplementary file 2* for insert sequence. |
| Recombinant DNA reagent | 1G4α | *Aleksic et al., 2010* | | Soluble 1G4 alpha chain for production in *E. coli* and in vitro refolding. |
| Recombinant DNA reagent | 1G4β | *Aleksic et al., 2010* | | Soluble 1G4 beta chain for production in *E. coli* and in vitro refolding. |
| Recombinant DNA reagent | 1G4β | *Abu-Shah et al., 2019* | | For mRNA electroporation. |
| Recombinant DNA reagent | 1G4α | *Abu-Shah et al., 2019* | | For mRNA electroporation. |
| Recombinant DNA reagent | CD3 ζ | *Abu-Shah et al., 2019* | | For mRNA electroporation. |
| Recombinant DNA reagent | HLA-A*02:01 heavy chain | *Aleksic et al., 2010* | | Soluble MHC heavy chain for production in *E. coli* and in vitro refolding. |
| Recombinant DNA reagent | β2M | *Aleksic et al., 2010* | | beta-2 microglobulin for production in *E. coli* and in vitro refolding. |
| Peptide, recombinant protein | Retronectin | Takara Bio | T100B | |
| Peptide, recombinant protein | Streptavidin-PE | Biolegend | 405245 | |
| Peptide, recombinant protein | Biotinylated BSA | Thermo Fisher Scientific | 29130 | |
| Peptide, recombinant protein | Streptavidin | Thermo Fisher Scientific | 434301 | |
| Peptide, recombinant protein | Peptide ligands | Peptide Protein Research | | See *Figure 1—source data* for details. |
| Peptide, recombinant protein | Refolded recombinant pMHCs | This paper | | Expressed in *E. coli*. |
| Peptide, recombinant protein | IL2 | PeproTech | 200-02 | |
| Peptide, recombinant protein | IL4 | PeproTech | 200-04 | |
| Peptide, recombinant protein | TNF | PeproTech | 300-01A | |
| Peptide, recombinant protein | IFNγ | R&D Systems | 285-IF-100/CF | |
| Peptide, recombinant protein | GM-CSF | Immunotools | 11343127 | |

*Continued on next page*

*Continued*

| Reagent type (species) or resource | Designation | Source or reference | Identifiers | Additional information |
|---|---|---|---|---|
| Peptide, recombinant protein | IL1β | R&D Systems | 201-LB-025/CF | |
| Commercial assay or kit | RosetteSep Human CD8+ T Cell Enrichment Cocktail | STEMCELL Technologies | 15063 | Isolation kits used to enrich for human immune cells from blood. |
| Commercial assay or kit | RosetteSep Human Monocyte Enrichment Cocktail | STEMCELL Technologies | 15068 | Isolation kits used to enrich for human immune cells from blood. |
| Commercial assay or kit | EasySep Human Memory CD8+ T Cell Enrichment Kit | STEMCELL Technologies | 19159 | Isolation kits used to enrich for human immune cells from blood. |
| Commercial assay or kit | EasySep Human NaÃ¯ve CD8+ T Cell Isolation Kit II | STEMCELL Technologies | 17968 | Isolation kits used to enrich for human immune cells from blood. |
| Commercial assay or kit | mMESSAGE mMACHINE T7 ULTRA Transcription Kit | Thermo Fisher Scientific | AM1345 | Prepare in vitro RNA transcripts. |
| Commercial assay or kit | MEGAclear Transcription Clean-Up Kit | Thermo Fisher Scientific | AM1908 | Isolate in vitro RNA transcripts. |
| Commercial assay or kit | Amine coupling kit | GE Healthcare Life Sciences | BR100050 | For immobilisation of protein on SPR chip. |
| Commercial assay or kit | IL-2 Human Uncoated ELISA Kit | Thermo Fisher Scientific | 88-7025-77 | |
| Chemical compound, drug | PGE2 | Sigma-Aldrich | P6532 | |
| Chemical compound, drug | Fixable Viability Dye eFluor 780 | Thermo Fisher Scientific | 65-0865-14 | |
| Chemical compound, drug | CD3/CD28 Human T-activator dynabeads | Thermo Fisher Scientific | 11132D | |
| Chemical compound, drug | X-tremeGENE HP | Sigma-Aldrich | 6366546001 | |
| Chemical compound, drug | Zombie Fixable viability kit | Biolegend | 423107; 423105 | Colours: UV, NIR; dilution: 1:1000. |
| Software, algorithm | Prism | GraphPad | | Data fitting and statistics. |
| Software, algorithm | Matlab | Mathworks | | Model fitting. |
| Software, algorithm | FlowJo | BD Biosciences | | |
| Other | 96 Well SensoPlate | Greiner | 655892 | |
| Other | Tetramers | This paper | | Made with NYE 9V or Tax WT and commercial streptavidin-PE. |
| Other | CM5 sensor chips | GE Healthcare Life Sciences | | |

## Protein production

Class I pMHCs were refolded as previously described (*Achour et al., 1999*). Human HLA-A*0201 heavy chain (UniProt residues 25–298) with a C-terminal AviTag/BirA recognition sequence and human beta-2 microgolublin were expressed in *Escherichia coli* and isolated from inclusion bodies. Trimer was refolded by consecutively adding peptide, β2M, and heavy chain into refolding buffer and incubating for 2–3 days at 4°C. Protein was filtered, concentrated using centrifugal filters, biotinylated (BirA biotin-protein ligase bulk reaction kit [Avidity, USA]), and purified by size exclusion chromatography

(Superdex 75 column [GE Healthcare]) in HBS-EP (0.01 M HEPES pH 7.4, 0.15 M NaCl, 3 mM EDTA, 0.005% v/v Tween20). Purified protein was aliquoted and stored at –80°C until use. Soluble α and β subunits of 1G4 and A6 TCRs were produced in *E. coli*, isolated from inclusion bodies, refolded in vitro, and purified using size exclusion chromatography in HBS-EP, as described previously (*Aleksic et al., 2010*).

Soluble extracellular domain (ECD) of human CD58 (UniProt residues 29–204 or 29–213) was produced either in Freestyle 293F suspension cells (Thermo Fisher) or adherent, stable GS CHO cell lines. For the latter, cells were expanded in selection medium (10% dialysed FCS, 1× GSEM supplement [Sigma-Aldrich], 20–50 μM MSX, 1% Pen/Strep) for at least 1 week. Production was performed in production medium (2–5% FCS, 1× GSEM supplement, 20 μM MSX, 2 mM sodium butyrate, 1% Pen/Strep) continuously for a few weeks with regular medium exchanges. Human ICAM1 ECD (UniProt residues 28–480) was either produced by transient transfection or lentiviral transduction of adherent 293T, or by transient expression in 293F. Production in 293F was performed according to the manufacturer's instructions using pTT3-ecdCD58 or pTT3-ecdICAM1. All supernatants were 0.45 μm filtered and 100 μM PMSF was added. Proteins were purified using standard Ni-NTA agarose columns, followed by in vitro biotinylation as described above. Alternatively, ligands were biotinylated by co-transfection (1:10) of a secreted BirA-encoding plasmid (pTT3-BirA-FLAG) and adding 100 μM D-biotin to the medium, as described before (*Parrott and Barry, 2001*). Proteins were further purified and excess biotin removed from proteins biotinylated in vitro by size exclusion chromatography (Superdex 75 or 200 column [GE Healthcare]) in HBS-EP; purified proteins were aliquoted and stored at –80°C until use.

Biotinylation levels of pMHC and accessory ligands were routinely tested by gel shift on SDS-PAGE upon addition of saturating amounts of streptavidin.

## Surface plasmon resonance

TCR–pMHC interactions were analysed on a Biacore T200 instrument (GE Healthcare Life Sciences) at 37°C and a flow rate of 10 μl/min. Running buffer was HBS-EP. Streptavidin was coupled to CM5 sensor chips using an amino coupling kit (GE Healthcare Life Sciences) to near saturation, typically 10,000–12,000 response units (RU). Biotinylated pMHCs (47 kDa) were injected into the experimental flow cells (FCs) for different lengths of time to produce desired immobilisation levels (typically 500–1500 RU), which were matched as closely as feasible in each chip. Usually, FC1 was as a reference for FC2–FC4. Biotinylated CD58 ECD (24 kDa + ~25 kDa glycosylation) was immobilised in FC1 at a level matching those of pMHCs. In some experiments, another FC was used as a reference. Excess streptavidin was blocked with two 40 s injections of 250 μM biotin (Avidity). Before injections of soluble 1G4 or A6 $\alpha\beta$ TCR (51 kDa), the chip surface was conditioned with eight injections of the running buffer. Dilution series of TCRs were injected simultaneously in all FCs; the duration of injections (30–70 s) was the same for conditioning and TCR injections. After every 2–3 TCR injections, buffer was injected to generate data for double referencing. After the final TCR injection and an additional buffer injection, W6/32 antibody (10 μg/ml; Biolegend; lot: B233942) was injected for 10 min.

TCR steady-state binding was measured >10 s post-injection. In addition to subtracting the signal from the reference FC with immobilised CD58 (single referencing), all TCR binding data was double referenced (*Myszka, 1999*) versus the average of the closest buffer injections before and after TCR injection. This allows to exclude small differences in signal between flow cells (e.g. drifts). TCR binding versus TCR concentration was fitted with the following model: $B = B_{max} * [TCR]/(K_D + [TCR])$, where $B$ is the response/binding, $B_{max}$ the maximal binding (this parameter is either kept free or is fixed with the W6/32-derived $B_{max}$), and $[TCR]$ the injected TCR concentration. Maximal W6/32 binding ($R_{max}$) was used to generate the empirical standard curve and to infer the $B_{max}$ of TCRs from the standard curve. $R_{max}$ was derived by fitting the W6/32 binding data after double referencing with the following, empirically chosen, model: $R = R_{max} * t/(K_t + t)$, where $t$ is time (s), $R$ the sensogram response after single referencing, and $K_t$ a nuisance parameter. The empirical standard curve only contained data where the ratio of the highest concentration of TCR to the fitted KD value (obtained using the standard method with $B_{max}$ fitted) was 2.5 or more. This threshold ensured that the binding response curves saturated so that only accurate measurements of $B_{max}$ were included. All interactions were fit using both the fitted and constrained Bmax method (*Figure 1E*). For constrained KD above 20 μM, we reported the constrained KD, otherwise we use the $B_{max}$ fitted KD. SPR data was analysed

using GraphPad Prism 8 (GraphPad software) or using a custom Python script (Python v3.7 and lmfit v0.9.13).

## Co-culture of naïve and memory T cells

The assay was performed as previously described (*Abu-Shah et al., 2019*). Naïve and memory T cells were isolated from anonymised HLA-A2$^+$ leukocyte cones obtained from the NHS Blood and Transplantation service at Oxford University Hospitals by (REC 11/H0711/7), using EasySep Human naïve CD8$^+$ T Cell Isolation Kit (STEMCELL) and EasySep Human Memory CD8$^+$ T Cell Enrichment Kit (STEMCELL), respectively. Cells were washed 3× with Opti-MEM serum-free medium (Thermo Fisher) and 2.5–5.0 Mio cells were resuspended at a density of 25 Mio/ml. Suspension was mixed with 5 µg/Mio cells of 1G4α, 1G4β, and CD3 $\zeta$ each, and 100–200 µl suspension was transferred into a BTX Cuvette Plus electroporation cuvette (2 mm gap; Harvard Bioscience). Electroporation was performed using a BTX ECM 830 Square Wave Electroporation System (Harvard Bioscience) at 300 V, 2 ms. T cells were used 24 hr after electroporation. 1G4 TCR contained an engineered cysteine (αT48C and βS57C) to reduce mispairing (*Cohen et al., 2007*).

Autologous monocytes were enriched from the same blood product using RosetteSep Human Monocyte Enrichment Cocktail (Stemcell), cultured at 1–2 Mio/ml in 12-well plates in the presence of 50 ng/ml IL4 (PeproTech) and 100 ng/ml GM-CSF (Immunotools) for 24 hr to induce differentiation. Maturation into moDCs was induced by adding 1 µM PGE$_2$ (Sigma Aldrich), 10 ng/ml IL1β (Biotechne), 20 ng/ml IFNγ, and 50 ng/ml TNF (PeproTech) for an additional 24 hr. MoDCs (50,000/well) were loaded for 60–90 min at 37°C with peptide and labelled with Cell Trace Violet (Thermo Fisher) to distinguish them from T cells prior to co-culturing with 50,000 T cells/well in a 96-well plate for 24 hr. T cell activation was assessed by flow cytometry and testing culture supernatant for cytokines using ELISAs.

## T cell blasts

All cell culture of human T cells was done using complete RPMI (10% FCS, 1% penicillin/streptomycin) at 37°C, 5% CO$_2$. T cells were isolated from whole blood from healthy donors or leukocyte cones purchased from the NHS Blood and Transplantation service at the John Radcliffe Hospital. For whole blood donations, a maximum of 50 ml was collected by a trained phlebotomist after informed consent had been given. This project has been approved by the Medical Sciences Inter-Divisional Research Ethics Committee of the University of Oxford (R51997/RE001), and all samples were anonymised in compliance with the Data Protection Act.

For plate stimulations and experiments with U87 target cells, CD8$^+$ T cells were isolated using RosetteSep Human CD8$^+$ enrichment cocktail (STEMCELL) at 6 µl/ml for whole blood or 150 µl/ml for leukocyte cones. After 20 min incubation at room temperature, blood cone samples were diluted 3.125-fold with PBS, while whole blood samples were used directly. Samples were layered on Ficoll Paque Plus (GE) at a 0.8:1.0 Ficoll:sample ratio and spun at 1200 g for 20–30 min at room temperature. Buffy coats were collected, washed twice, counted, and cells were resuspended in complete RMPI with 50 U/ml IL2 (PeproTech) and CD3/CD28 Human T-activator dynabeads (Thermo Fisher) at a 1:1 bead:cell ratio. Aliquots of 1 Mio cells in 1 ml medium were grown overnight in 12- or 24-well plates (either TC-treated or coated with 5 µg/cm$^2$ retronectin [Takara Bio]) and then transduced with VSV-pseudotyped lentivirus encoding for either the 1G4 or the A6 TCR. After 2 days (4 days after transduction), 1 ml of medium was exchanged, and IL2 was added to a final concentration of 50 U/ml. Beads were magnetically removed at day 5 post-transduction, and T cells from thereon were resuspended at 1 Mio/ml with 50 U/ml IL2 every other day. For functional experiments, T cells were used between 10 and 16 days after transduction.

## Lentivirus production

HEK 293T or Lenti-X 293T (Takara) were seeded in complete DMEM in 6-well plate to reach 60–80% confluency after 1 day. Cells were either transfected with 0.95 µg pRSV-Rev, 0.37 µg pVSV-G (pMD2.G), 0.95 µg pGAG (pMDLg/pRRE), and 0.8 µg of pLEX-A6 or pHR-1G4 with 9 µl X-tremeGENE nine or HP (both Roche). Lentiviral supernatant was harvested after 20–30 hr and filtered through a 0.45 µm cellulose acetate filter. In an updated version, LentiX cells were transfected with 0.25 µg pRSV-Rev, 0.53 µg pGAG, 0.35 µg pVSV-G, and 0.8 µg transfer plasmid using 5.8 µl X-tremeGENE HP. Medium was

replaced after 12–18 hr, and supernatant harvested as above after 30–40 hr. Supernatant from one well of a 6-well plate was used to transduce 1 Mio T cells. Sequence for the A6 TCR lacked one natural cysteine per chain and included engineered cysteines (αT48C and βS57C) to reduce the formation of mixed TCR dimers with endogenous TCR (*Cohen et al., 2007*). The 1G4 TCR was expressed from the WT sequences without engineered cysteines.

## Co-culture of T cell blasts

For co-culture experiments with U87 (a kind gift of Vincenzo Cerundolo, University of Oxford), 30,000 target cells were seeded in a TC-coated 96-well F-bottom plate and incubated overnight. Peptides were diluted in complete DMEM (10% FCS, 1% penicillin/streptomycin) to their final concentration and incubated with U87 cells for 1–2 hr at 37°C. Peptide-containing medium was removed and 60,000 TCR-transduced primary human CD8⁺ T cell blasts were added, spun for 2 min at 50 g, and incubated for 5 hr at 37 °C. At the end of the experiment, 10 mM EDTA was added and cells were detached by vigorous pipetting. Cells were stained for flow cytometry and analysed immediately, or fixed and stored for up to 1 day before running. Supernatants were saved for cytokine ELISAs.

## Plate stimulation

Glass-bottom Sensoplates (96-well; Greiner) were washed with 1 M HCl/70% EtOH, thoroughly rinsed twice with PBS, and coated overnight at 4°C with 100 µl/well of 1 mg/ml biotinylated BSA (Thermo Fisher) in PBS. Plates were washed with PBS twice and incubated for at least 1 hr with 20 µg/ml streptavidin (Thermo Fisher) in 1% BSA/PBS at room temperature. Plates were washed again with PBS and biotinylated pMHC (in-house) was added for at least 1 hr at room temperature or overnight at 4°C. Plates were emptied and accessory ligand (CD58 or ICAM1, in-house) or PBS was added for the same duration as above. Upon completion, plates were washed once and stored for up to 1 day in PBS at 4°C.

For stimulation, T cells were counted, washed once to remove excess IL2, and 75,000 cells in 180–200 µl complete RMPI were dispensed per well. Cells were briefly spun down at 50 g to settle to the bottom and subsequently incubated for 4 hr at 37°C. At the end of the experiment, 10 mM EDTA was added and cells were detached by vigorous pipetting. Cells were stained for flow cytometry and analysed immediately, or fixed and stored for up to 1 day. Supernatants were saved for cytokine ELISAs.

## Peptides and loading

We used peptide ligands that were either described previously (*Aleksic et al., 2010*; *Lever et al., 2016*; *Ding et al., 1998*; *Ding et al., 1999*; *Gagnon et al., 2006*; *Borbulevych et al., 2009*; *Borbulevych et al., 2011*) or designed by us based on the published crystal structures of these TCRs in complex with MHC (1G4: PDB 2BNQ, A6: PDB 1AO7).

Peptides were synthesised at a purity of >95% (Peptide Protein Research, UK). Tax WT is a 9 amino acid, class I peptide derived from HTLV-1 Tax$_{11-19}$ (*Utz et al., 1996*; *Garboczi et al., 1996*). NYE 9V refers to a heteroclitic (improved stability on MHC), 9 amino acid, class I peptide derived from the wild-type NYE-ESO$_{157-165}$ 9C peptide (*Chen et al., 2005*). See *Figure 1—source data* for a list of peptides.

Loading efficiency was evaluated by pulsing T2 cells for 1–2 hr at 37°C with a titration of peptides. Loading was assessed as upregulation of HLA-A2 (clone: BB7.2; Biolegend) by flow cytometry.

## Flow cytometry

Tetramers were produced in-house using refolded monomeric, biotinylated pMHC, and streptavidin-PE (Biolegend) at a 1:4 molar ratio. Streptavidin-PE was added in 10 steps and incubated for 10 min while shaking at room temperature. Insoluble proteins were removed by brief centrifugation at 13,000 g and 0.05–0.1% sodium azide added for preservation. Tetramers were kept for up to 3 months at 4°C. Cells were stained for CD69 with clones FN50 (Biolegend). Staining for CD45 (clone HI30; Biolegend) was used to distinguish target and effector cells in co-culture assays with U87 cells. Cell viability staining was routinely performed for plate stimulations and U87 co-culture using fixable violet or near-infrared viability dyes (Zombie UV fixable viability kit [Biolegend], Zombie NIR fixable viability

kit [Biolegend], eBioscience fixable viability dye eFluor 780 [Invitrogen]). Samples were analysed using a BD X-20 flow cytometer, and data analysis was performed using FlowJo v10 (BD Biosciences).

### ELISAs

Human IL-2 Ready-SET Go! ELISA kit (eBioscience/Invitrogen) or Human TNF alpha ELISA Ready-SET-Go! (eBioscience/Invitrogen) and Nunc MaxiSorp 96-well plates (Thermo Fisher) were used according to the manufacturer's instructions to test appropriately diluted (commonly 4–30-fold) T cell supernatant for secretion of IL2 or TNF.

### TCR expression

TCR $\alpha\beta$- KO Jurkat E6.1 cells (a kind gift of Edward Jenkins) were transduced with 1G4 or A6 lentivirus, and TCR expression was measured by staining for CD3 (clone: UCHT1; Biolegend) and TCR $\alpha\beta$ (clone IP26; Biolegend).

### Data analysis

Quantitative analysis of antigen discrimination was performed by first fitting dose-response data with a four-parameter sigmoidal model on a linear scale in Python v3.7 and lmfit v0.9.13 using Levenberg–Marquardt:

$$R(x) = E_{min} + \frac{E_{max} - E_{min}}{1 + (\frac{EC_{50}}{x})^H}$$

where $x$ refers to the peptide concentration used to pulse the target cells (in μM) or the amount of pMHC used to coat the well of a plate (in ng/well). The curve produced by this fit was used to interpolate potency as the concentration of antigen required to induce activation of 15% for CD69 ($P_{15}$) and 10% for IL2 ($P_{10}$). These percentages were chosen based on noise levels and to include lower-affinity antigens in the potency plots. Potency values exceeding doses used for pulsing or coating were excluded from the analysis (i.e. no extrapolated data was included in the analysis).

To determine the discrimination power α, we fitted the power law in log-space to our data:

$$P'_{15} = C + \alpha K'_D$$

where $P'_{15} = \log_{10}(P_{15})$ and $K'_D = \log_{10}(K_D)$. All data analysis was performed using GraphPad Prism (GraphPad Software), if not stated otherwise.

### Kinetic proofreading: fitting to data

#### Deriving the expression for ligand potency

A pMHC ligand $L$ can bind with a TCR $R$ to create a complex $C_0$ at a rate $k_{on}$. In order for this complex to initiate TCR signalling, it undergoes a series of $N$ steps. We denote by $C_i$ a TCR/pMHC complex in the ith KP step. A complex $C_i$ becomes a complex $C_{i+1}$ with rate $k_p$, for $0 \leq i \leq N - 1$. At any KP step the pMHC ligand can unbind with rate $k_{off}$. Let $L(t)$, $R(t)$, and $C_i(t)$ be the concentration of ligand, receptor, and complex in the ith KP step at time $t$, respectively. The system of ordinary differential equations that govern the temporal evolution of the concentrations is given by

$$\frac{dL(t)}{dt} = k_{off} \sum_{i=0}^{N} C_i(t) - k_{on}L(t)R(t) \tag{1a}$$

$$\frac{dR(t)}{dt} = k_{off} \sum_{i=0}^{N} C_i(t) - k_{on}L(t)R(t) \tag{1b}$$

$$\frac{dC_0(t)}{dt} = k_{on}L(t)R(t) - (k_{off} + k_p) C_0(t) \tag{1c}$$

$$\frac{dC_i(t)}{dt} = k_p C_{i-1}(t) - (k_{off} + k_p) C_i(t), \qquad \text{for } 1 \leq i \leq N - 1, \tag{1d}$$

$$\frac{dC_0(t)}{dt} = k_{on}L(t)R(t) - (k_{off} + k_p) C_0(t). \tag{1e}$$

Let the initial number of pMHC ligands and TCRs be $L_0$ and $R_0$, respectively. We then define the total number of complexes at time $t$ as $C_{\text{tot}}(t) = \sum_{i=0}^{N} C_i(t)$ and note the two conservation equations, $L_0 = L(t) + C_{\text{tot}}(t)$ and $R_0 = R(t) + C_{\text{tot}}(t)$. Solving the steady-state equations arising from setting the time derivatives in *Equation 1* to zero, and substituting in the conservation equations we find that

$$C_N = \left(1 + \frac{k_{\text{off}}}{k_p}\right)^{-N} C_{\text{tot}},\tag{2}$$

where

$$C_{\text{tot}} = \frac{L_0 + R_0 + \frac{k_{\text{off}}}{k_{\text{on}}} - \sqrt{\left(L_0 + R_0 + \frac{k_{\text{off}}}{k_{\text{on}}}\right)^2 - 4L_0 R_0}}{2}.\tag{3}$$

The expression in *Equation 2* determines the concentration of actively signalling TCR/pMHC complexes $C_N$ for a given number of ligands $L_0$. To fit this model to the potency data, we are interested in calculating the concentration of pMHC ligand required to initiate T cell activation for different TCR/pMHC binding parameters. We first introduce a few convenient rescalings and redefinitions. We define $x = L_0/R_0$ to be the potency of ligand concentration relative to the total number of receptors and let $\lambda = C_N/R_0$ be a threshold parameter that dictates how much $C_N$ complex is needed to activate a T cell response relative to the total number of receptors. Thus *Equation 2* can be rewritten as

$$2\lambda \left(1 + \frac{k_{\text{off}}}{k_p}\right)^N = 1 + x + \frac{k_{\text{off}}}{R_0 k_{\text{on}}} - \sqrt{\left(1 + x + \frac{k_{\text{off}}}{R_0 k_{\text{on}}}\right)^2 - 4x}.\tag{4}$$

The experimental measurements of potency do not directly correspond to the potency $x$ in our model as the exact number of ligand and receptor is unknown. Therefore, we introduce a constant of proportionality $\gamma$ into our model such that $x \to \gamma x$. Similarly, the ratio $k_{\text{off}}/k_{\text{on}}$ is a measure of ligand affinity and is directly proportional to the experimental KD values, thus we introduce a second constant of proportionality $\delta$ such that $k_{\text{off}}/(R_0 k_{\text{on}}) \to \delta K_D$, where we absorb the constant $R_0$ into the new parameter. With these adjustments, *Equation 4* becomes

$$2\lambda \left(1 + \frac{k_{\text{off}}}{k_p}\right)^N = 1 + \gamma x + \delta K_D - \sqrt{\left(1 + \gamma x + \delta K_D\right)^2 - 4\gamma x}.\tag{5}$$

Upon rearranging *Equation 5*, we find that

$$-\sqrt{\left(1 + \gamma x + \delta K_D\right)^2 - 4\gamma x} = 2\lambda \left(1 + \frac{k_{\text{off}}}{k_p}\right)^N - \left(1 + \gamma x + \delta K_D\right),\tag{6}$$

we then square (squaring both sides will not introduce a false solution so long as $\lambda \left(1 + k_{\text{off}} k_p\right)^N < 1$) both sides of *Equation 6* and find the following expression for the potency:

$$x = \frac{\lambda \left(1 + \frac{k_{\text{off}}}{k_p}\right)^N}{\gamma} \left[1 - \frac{\delta K_D}{\lambda \left(1 + \frac{k_{\text{off}}}{k_p}\right)^N - 1}\right].\tag{7}$$

## Fitting the potency expression using ABC-SMC parameter estimation

We used the Approximate Bayesian Computation-Sequential Monte Carlo (ABC-SMC) algorithm to determine the distribution of KP model parameters that fit the experimental data. Our KP model has five parameters, $N$, $k_p$, $\lambda$, $\gamma$, and $\delta$. We fit the model parameters to the plate and the cell data separately. For both the plate and the cell data, we fit $N$, $\gamma$, and $\delta$ as a global parameter shared amongst all experimental repeats. The parameters $k_p$ and $\lambda$ are fitted locally for each repeat. We fit the potency equation to the experimental data in log space, and as such the log expression for potency, $\rho\left(N, k_p, \hat{\lambda}, \gamma, \hat{\delta}\right)$, calculated from *Equation 7* is given by

$$\rho\left(N, k_p, \hat{\lambda}, \gamma, \hat{\delta}; K_D\right) = \log_{10}\left(\hat{\lambda}\right) + N \log_{10}\left(1 + \frac{k_{off}}{k_p}\right) + \log_{10}\left(1 - \frac{\hat{\delta} K_D}{\hat{\lambda}\left(1 + \frac{k_{\text{off}}}{k_p}\right)^N - \frac{1}{\gamma}}\right),\tag{8}$$

where $\hat{\lambda} = \lambda/\gamma$ and $\hat{\delta} = \delta/\gamma$. These rescalings ensure that the parameters are orthogonal and thus parameter space can be searched efficiently. The fast kinetics of the low-affinity pMHCs precluded direct measurements of $k_{off}$, and instead, we noted that on-rates exhibit small variations between pMHCs that differ by few amino acids (*Aleksic et al., 2010*; *Lever et al., 2016*). Therefore, we estimated $k_{off}$ using $K_D$ and a fixed $k_{on}$ of 0.0447 µM-1s-1 taken as the average $k_{on}$ of NYE 9C, 9V, 3A, 3I, 3M, 3Y, and 6V previously measured at 37°C (*Aleksic et al., 2010*).

We chose uniform prior distributions in log space for each parameter except $N$, where a uniform prior in linear space was used. This allows for efficient search through parameter space over many orders of magnitude. The priors for the plate data are as follows:

$$N \sim \mathrm{Unif}\,(0, 4),\tag{9a}$$

$$\log_{10}\,(k_p) \sim \mathrm{Unif}\,(-1, 1),\tag{9b}$$

$$\log_{10}\,\left(\hat{\lambda}\right) \sim \mathrm{Unif}\,(-4, 1),\tag{9c}$$

$$\log_{10}\,(\gamma) \sim \mathrm{Unif}\,(-6, -4),\tag{9d}$$

$$\log_{10}\,\left(\hat{\delta}\right) \sim \mathrm{Unif}\,(-7, -5),\tag{9e}$$

where the priors for the cell data are the same other than for $\hat{\lambda}$ where $\log_{10}\,\left(\hat{\lambda}\right) \sim \mathrm{Unif}\,(-6, -3)$.

Recall that we fit the parameters $N$, $\gamma$, and $\hat{\delta}$ globally and $\hat{\lambda}$ and $k_p$ are fitted locally. For the plate data, this results in 27 parameters to fit whilst for the cell data there are 37 parameters. Let $\Theta = \left(N, \gamma, \hat{\delta}, \vec{k_p}, \vec{\hat{\lambda}}\right)$ be the vector of parameters to fit such that the ith entry of the vectors $\vec{k_p}$ and $\vec{\hat{\lambda}}$ correspond to the local parameters for the ith experiment. Then let $\vec{K_D}^{\,i}$ be the vector of experimentally measured KD values, and $\vec{P}^i$ be the vector of potency measurements for the ith experiment. These vectors differ in length and so we denote by $d_i$ the number of data points in the ith experiment. We measure the similarity between the KP model and the experimental results via the following distance function:

$$\mathcal{D}\,(\Theta) = \sum_{i=1}^{I} \sum_{j=1}^{d_i} \left(\rho\left(N, \left[\vec{k_p}\right]_i, \hat{\lambda}_i, \gamma, \hat{\delta}; \left[\vec{K_D}^{\,i}\right]_j\right) - \log_{10}\left(\left[\vec{P}^i\right]_j\right)\right)^2,\tag{10}$$

where $I$ denotes the total number of experiments, $I = 12$ and $I = 17$ for the plate and cell data, respectively.

To perform a randomised search through the parameter space, we employed the following Metropolis–Hastings algorithm. We sample an initial parameter set $\Theta_0$ from the prior distributions detailed above. Let $\Theta_{curr}$ denote the current set of parameters which initially is $\Theta_0$. A candidate set of parameters, $\Theta_{cand}$, is found by adding a random perturbation to $\Theta_{curr}$. The perturbation is achieved by adding a uniform random shift to each parameter in $\Theta_{curr}$ independently. The range of the uniform random shift is $[-0.005, 0.005]$ multiplied by the width of the prior. For example, we perturb the $N$ parameter by adding a random uniform shift in the interval $[-0.02, 0.02]$. If the parameter falls outside the bounds in the prior distribution, it is reflected symmetrically back within the bounds. We then have to decide whether to accept or reject the candidate set of parameters. If $\mathcal{D}\,(\Theta_{cand}) < \mathcal{D}\,(\Theta_{curr})$, then we accept the parameters as they share a greater similarity with the experimental data and set $\Theta_{curr} = \Theta_{cand}$. Otherwise we only accept the candidate parameters with probability $\exp\left(-\left(\mathcal{D}\,(\Theta_{cand}) - \mathcal{D}\,(\Theta_{curr})\right)/\xi\right)$, where $\xi$ is a parameter that controls how likely accepting a set of parameters with a higher distance function is. The value of $\xi$ is reduced as the algorithm gets closer to a set of parameters that minimises the distance function. Initially $\xi = 10$ but is subsequently reduced to $\{1, 0.1, 0.01, 0.005, 0.001\}$ when the distance function of the candidate set of parameters first reaches $\{50, 30, 20, 18, 17.5\}$ for the plate data and $\{100, 75, 50, 40, 35\}$ for the cell data. The algorithm continues until it reaches a final set of parameters that has a distance less than 11.08 or 39.2 for the plate and cell data, respectively. For both the plate and cell data, we performed this algorithm 1000 times to capture the distribution of parameter values that fit the experimental data.

The ABC-SMC algorithm described above was implemented with custom C++ code (Apple LLVM version 7.0.0, clang-700.1.76). The distributions of the parameters are presented in *Figure 5—figure supplement 1*.

## Kinetic proofreading: binary heatmaps of discrimination and sensitivity

We defined measures of sensitivity and discrimination in order to test whether the KP mechanism can explain both for different KP model parameters. Recall that $\lambda$ is the minimum threshold concentration of productively signalling TCR/pMHC complexes in the Nth step. To determine TCR sensitivity, we require that the number of productively signalling TCRs is above the threshold for a single agonist pMHC with the highest affinity $K_{D;1} = k_{\text{off};1}/k_{\text{on}}$. From **Equation 3**, we can make the approximation $C_{\text{total}} \approx \min\left(L_0, R_0\right)$ when $L_0 + R_0 \gg K_{D;1}$. Then, noting that $\min\left(1, R_0\right) = 1$ and using **Equation 2** we can write the sensitivity requirement as the following inequality:

$$C_N = \left(1 + \frac{k_{\text{on}} K_{D;1}}{k_p}\right)^{-N} > \lambda. \tag{11}$$

To determine TCR discrimination, we determined whether the number of productively signalling TCRs was below the same threshold $\lambda$ for a pMHC that was expressed at 10,000-fold higher concentration but bound with a $\Delta$-fold lower affinity. With our empirical equation for the discrimination power ($P = 10^C K_D^\alpha$), we can calculate the potency $P$ for a given ligand affinity. Assuming $K_D$ is proportional to $k_{\text{off}}$ and $P$ is a ligand concentration needed to activate the TCR $L_0$, we can rewrite the equation as $L_0 = 10^C k_{\text{off}}^\alpha$. The difference in potency between the ligand interaction with the higher affinity $K_{D;1}$ and a ligand with lower affinity $K_{D;2}$ is hence:

$$\frac{L_{0;2}}{L_{0;1}} = \left(\frac{k_{\text{off};2}}{k_{\text{off};1}}\right)^\alpha \tag{12}$$

As we require $L_{0;1}$ to be 1 to fulfil the sensitivity constrain, the equation simplifies to $L_{0;2} = \Delta^\alpha$ with $\Delta$ being the difference in affinity between the two ligands. Hence, a ligand with $\Delta$-fold lower-affinity than the higher-affinity ligand will need a concentration of $L_{0;2}$ ligands for activation. For the discrimination constraint, we require that a ligand with $\Delta$-fold lower affinity than the highest affinity ligand needs $L_{0;2}$ or more ligands to overcome the threshold of activation. The discrimination requirement can be written as the following inequality:

$$C_N = \min\left(L_{0;2}, R_0\right)\left(1 + \Delta \frac{k_{\text{on}} K_{D;1}}{k_p}\right)^{-N} < \lambda. \tag{13}$$

Both of these constraints must be fulfilled simultaneously for a given set of KP parameters in order for the kinetic proofreading model to explain both sensitivity and discrimination.

For the simulation of the KP model (**Figure 5G–I**), we choose $\Delta$ such that $L_{0;2} = 10,000$ according to $\Delta_A = 10000^{1/\alpha}$. Given that the number of TCRs is $R_0 \sim 30,000$, choosing $\Delta_{L0}$ means that the receptors are not saturated with ligands and potency varies linearly with affinity. The final discrimination constraint function is as follows:

$$C_N = 10000(1 + 10000^{1/\alpha} \frac{k_{\text{on}} K_{D;1}}{k_p})^{-N} < \lambda. \tag{14}$$

In addition to using the deterministic KP model, we also calculated these sensitivity and discrimination measures using discrete stochastic simulations. We varied $N$ and $\tau = 1/k_p$. For each pair of parameters $(N, \tau)$, we simulate 250 realisations of the kinetic proofreading model using a standard Gillespie algorithm until a termination time of $t = 100$ s, which is sufficient in order for the model to have reached a steady state. From this ensemble, an average number of receptors in the final (Nth) proofreading step, $\langle C_N \rangle$, is calculated. This ensemble average is compared to the threshold for activation $\lambda = 0.1$.

Testing for both sensitivity and discrimination for each parameter pair $(N, \tau)$ requires simulating the model in two different scenarios. The first scenario is with a single ligand and unit dissociation rate, that is, $k_{\text{off}} = 1$. If the ensemble average $\langle C_N \rangle > 0.1$, then the parameter pair $(N, \tau)$ observes sensitivity and is shown as a red asterisk in the panels in **Figure 5G–I**. For discrimination, we increase the number of ligands to $\Delta_L = 10000$ and decreased the affinity of the ligand by $\Delta_A = 10000^{1/\alpha}$, that is, $k_{\text{off}} = 10000^{1/\alpha}$. If the average number of receptors $\langle C_N \rangle < 0.1$, then discrimination is observed, and the parameter pair $(N, \tau)$ is shown as a blue square in **Figure 5G–I**. Parameter pairs that are shown with

both a red asterisk and a blue square observe both sensitivity and discrimination. All stochastic simulations were performed with custom Julia code using the package *DifferentialEquations.jl*.

## Analysis of the discriminatory power for TCRs from published studies

*Figure 3—source data 1* provides information on each calculation of α and specific details on the source of data underlying each calculation (see Appendix 1).

The broad method was to obtain a measure of ligand potency from each study. If provided by the study, this was often an $EC_{50}$, which is the concentration of ligand eliciting 50% of the maximum response. If not explicitly provided, we estimated ligand potency as $P_X$, which was defined by the concentration of ligand that produced $X$ response. To do this, we drew a horizontal line at $X$ on a provided dose-response graph and estimated the ligand concentration where the data intercepted the horizontal line. The disadvantage with this method is that ligand potency was estimated based on the single representative graph provided in the study.

Each study often contained or cited a study that contained estimates of $K_D$ or $k_{off}$ for the specific TCR/pMHC interactions used in the study. We only included studies where monomeric SPR binding data was available to avoid multimeric binding parameters (e.g. when using tetramers). However, when analysing discrimination by other non-TCR receptors, we included binding data from various methods (e.g. SPR, radio labelled ligands) provided they were monomeric measurements. The use of SPR is important for weak interactions, such as TCR/pMHC, but various methods are available for higher-affinity interactions.

The plot of potency over $K_D$ or $k_{off}$ was fit using linear regression on log-transformed axes. We reported the slope of the fit (i.e. the discrimination power, α), the goodness-of-fit measure ($R^2$), and the p-value for the null hypothesis that the slope is zero (i.e. $\alpha = 0$). We defined significance using the threshold of p=0.05. We found that the calculated α was robust to the precise definition of ligand potency so that the same slope was produced when using a different response threshold (e.g. 0.25 or 0.75 instead of the commonly used value of 0.5, not shown).

A subset of the data relied on engineered high-affinity TCR/pMHC interactions. It has been observed that increasing the affinity beyond a threshold does not improve ligand potency (*Irving et al., 2012*; *Lever et al., 2016*). To avoid underestimating the discrimination power, we found that globally removing data where $K_D < 1$ μM avoided entering this saturation regime (with a single exception, see ID 58–61 in Supplementary information and *Figure 3—figure supplement 3*). Similarly, to avoid over-estimating α, we did not include data where the potency was extrapolated (i.e. when $EC_{50}$ values were larger than the highest ligand concentration tested). Some studies provided multiple measures of T cell responses, and in this case, we produced potency plots for each response and hence were able to obtain multiple estimates of α.

We only included discrimination powers in final comparisons (*Figures 3I–O and 6F*) that were statistically significant ($p < 0.05$) with the exception of the original and revised mouse TCR data (*Figure 3I*) because only few data were available. We found more studies that performed functional experiments on the original mouse TCRs compared to those that measured binding, and therefore to avoid introducing a potential bias in the analysis, we included only a single calculated α for each independent SPR measurement. In the case of the original mouse TCR data, we included four calculations of α (*Figure 3—source data 1*, ID 1, 2, 11, 14), and in the case of the revised mouse TCR data, we included six calculations of α (*Figure 3—source data 1*, ID 5, 13, 15, 17, 18, 19). We also note that discrimination powers obtained using artificial conditions, when antigen was presented on plates as recombinant protein or when presented on APCs but co-receptors were blocked, were *not* included in aggregated analyses (*Figures 3I–O and 6F*).

## Analysis of the discriminatory power for other surface receptors from published studies

*Figure 6—source data 1* provides information on each calculation of α and specific details on the source of data underlying each calculation (see Appendix 1).

The general method was similar to that used for the TCR (see previous section). We provide specific information on the analysis of each receptor family below.

Cytokine receptors transduce signals by ligand-induced dimerisation of receptor subunits. We identified five studies that produced ligands with mutations that modified binding to either one or

both receptor subunits and either reported potency or provided dose-response curves from which potency can be extracted (*Levin et al., 2012*; *Moraga et al., 2015*; *Thomas et al., 2011a*; *Mendoza et al., 2017*; *Martinez-Fabregas et al., 2019*). As an example, Moraga et al. generated IL-13 variants with mutations that resulted in a broad range of affinities to the IL-13Rα1 subunit but maintained the wild-type interface, and hence the same affinity, to the IL-4Rα subunit (*Moraga et al., 2015*). By measuring cellular responses, such as upregulation of CD86 on monocytes, dose-response curves were generated for each IL-13 variant, allowing us to determine ligand potency. We observed a significant correlation between potency and $K_D$ (*Figure 6A*). We repeated the analysis for each study (*Figure 6—source data 1* ID 1–13). In studies that included ligands with mutations to both receptor interfaces, we plotted potency over the product of the dissociation constants to each interface since this serves as an estimate of the overall affinity (i.e. $K_D^1 \times K_D^2$). Collating these studies revealed a mean discrimination power of $\alpha = 0.66$ (*Figure 6F*).

Like cytokine receptors, RTKs transduce signals by ligand-induced dimerisation. We identified two potential studies to include in the analysis (*Ho et al., 2017*; *Reddy et al., 1996*). Ho et al. generated stem cell factor (SCF) ligand variants to the RTK c-Kit (*Ho et al., 2017*). SCF induces c-Kit dimersation by binding to c-Kit with one interface and binding to another SCF with a different interface generating SCF/c-Kit homodimers. Four SCF variants were used in detailed dose-response assays measuring phosphorylation of ERK (*Figure 6B*, left) and AKT (not shown here). Given that the SCF variants included mutations impacting both c-Kit binding and SCF homodimerisation, we plotted potency over the product of the dissociation constants for each interface finding a significant correlation for ERK (*Figure 6B*, right) and AKT (*Figure 6—figure supplement 1* ID 16) with discrimination powers of 0.83 and 0.88, respectively. A significant correlation was not observed for the second study using EGFR (*Figure 6—source data 1* ID 14), and therefore, we estimated the mean for RTK based on the c-Kit data to be $\alpha = 0.86$ (*Figure 6F*).

Although multiple ligands for a given GPCR have been described, they often bind at different GPCR sites to stabilise different receptor conformations and hence transduce qualitatively different signals. Therefore, ligand affinity may not correlate to functional potency. Instead, we focused on identifying studies that used ligands that were confirmed to bind to the same interface with different affinities. As an example, Sykes et al. used seven agonists to the muscarinic $M_3$ receptor and confirmed they all bound to the same interface using a binding competition assay (*Sykes et al., 2009*). Using titrations of each ligand, they examined the binding of GTPγS to CHO-$M_3$ membrane as a measure of response (*Figure 6C*, left). Plotting ligand agonist potency over $K_D$ produced a significant correlation with a discrimination power of $\alpha = 0.55$ (*Figure 6C*, right). We found a similar discrimination power when using a different measure of response (Ca2+ mobilisation from CHO-$M_3$ cells) from the same study, and moreover, similar discrimination powers in other studies investigating the A2A receptor (*Guo et al., 2012*) and the chemokine receptors CXCR4 (*Guyon et al., 2013*) and CXCR3 (*Heise et al., 2005*; *Figure 6—source data 1* ID 17–24). Collating these studies revealed a mean discrimination power of $\alpha = 0.76$ (*Figure 6F*).

Chimeric antigen receptors (CARs) are therapeutic receptors often expressed in T cells that fuse an extracellular antigen recognition domain to an intracellular signalling domain (often the $\zeta$-chain from the TCR). Chmielewski et al. generated a panel of CARs that bind the ErbB2 receptor (target antigen) with different affinities (*Chmielewski et al., 2004*). CAR-T cells were stimulated with a titration of recombinant ErbB2, and their ability to produce the cytokine IFNγ was used to measure T cell responses (*Figure 6D*, left). We found a significant correlation between potency and $K_D$ with a discrimination power of $\alpha = 0.52$ (*Figure 6D*, right). Similar results were observed using a different ErbB2 CAR (*Liu et al., 2015*) and a DNA-based CAR (*Taylor et al., 2017*; *Figure 6—source data 1* ID 25–28). Together, we found a mean discrimination power of $\alpha = 0.94$ (*Figure 6F*).

Lastly, antigen discrimination has also been reported for the BCR, which shares many structural and functional features with the TCR. Although several studies have investigated BCR ligand discrimination, we identified only a single study with the requisite dose-response curves to quantify discrimination. Batista et al. used two lysozyme-specific BCRs (HyHEL10 and D1.3) to perform dose-response curves to wild-type or mutated lysozyme variants measuring the production of the cytokine IL-2 (*Figure 6E*, left; *Batista and Neuberger, 1998*). We estimated potency directly from the dose-response curves and found a significant correlation with $k_{off}$ (*Figure 6E*, right). We found the discrimination power for both HyHEL10 and D1.3 BCRs to be > 1 (mean of $\alpha = 1.3$, *Figure 6F*).

## Statistical analyses

All statistics on discrimination power and sensitivity were performed on log-transformed data, unless stated otherwise. In *Figure 1F*, data was compared using a Kruskal–Wallis test with Dunn's multiple comparison. In *Figure 2K*, conditions were compared with a one-way ANOVA and each condition was compared to $\alpha = 1$ with an independent one-sample Student's t-test; and in *Figure 2L*, the 1G4 data was compared with an ordinary one-way ANOVA and all data was compared using a second ordinary one-way ANOVA with Sidak's multiple comparison for a pairwise test. In *Figure 3*, all comparisons were performed using parametric one-way ANOVA and/or multiple t-tests (with the stated correction for multiple comparisons) on log-transformed data. In *Figure 4F*, plate data was compared using repeated-measure one-way ANOVA (Geisser–Greenhouse corrected) with Sidak's comparison for the indicated pairwise comparison. CD58 and ICAM1 were compared to U87 co-culture data using ordinary one-way ANOVA. Each condition was compared to $\alpha = 1$ using an independent one-sample Student's t-test. In *Figure 4G*, comparison using repeated-measure one-way ANOVA (Geisser–Greenhouse corrected). In *Figure 5D*, plate data compared using a repeated-measure one-way ANOVA (Geisser–Greenhouse corrected) and APC data and APC vs. plate data was compared using each an ordinary one-way ANOVA. In *Figure 6F*, a one-way ANOVA compares other receptors and a t-test compares other receptors to the TCR.

## Acknowledgements

We thank Ignacio Moraga Gonzalez, David K Cole, David R Greaves, Philipp Krüger, Edward Jenkins, Marcus Bridge, Samuel A Isaacson, Marion H Brown, and Tal Arnon for helpful discussions.

## Additional information

### Competing interests

Michael L Dustin: Reviewing editor, *eLife*. The other authors declare that no competing interests exist.

### Funding

| Funder | Grant reference number | Author |
| --- | --- | --- |
| Wellcome Trust | 207537/Z/17/Z | Johannes Pettmann<br>Anna Huhn<br>Enas Abu Shah<br>Mikhail A Kutuzov<br>Daniel B Wilson<br>Omer Dushek |
| Wellcome Trust | 098274/Z/12/Z | Simon J Davis |
| Wellcome Trust | 100262Z/12/Z | Michael L Dustin |
| Wellcome Trust | 203737/Z/16/Z | Johannes Pettmann |
| University of Oxford | Oxford-UCB Post-doctoral Fellowship | Enas Abu Shah |
| National Science Foundation | NSF-DMS 1902854 | Daniel B Wilson |
| Edward Penley Abraham Trust Studentship | | Anna Huhn |

The funders had no role in study design, data collection and interpretation, or the decision to submit the work for publication.

### Author contributions

Johannes Pettmann, Conceptualization, Data curation, Formal analysis, Funding acquisition, Investigation, Project administration, Visualization, Writing – original draft, Writing – review and editing; Anna Huhn, Data curation, Formal analysis, Funding acquisition, Investigation, methodology, Visualization, Writing – original draft, Writing – review and editing; Enas Abu Shah, Conceptualization, Data

curation, Funding acquisition, Investigation, methodology, Project administration, Writing – review and editing; Mikhail A Kutuzov, Conceptualization, Investigation, methodology, resources, Writing – review and editing; Daniel B Wilson, Data curation, Formal analysis, Investigation, Writing – review and editing; Michael L Dustin, Simon J Davis, Funding acquisition, supervision, Writing – review and editing; P Anton van der Merwe, Conceptualization, Funding acquisition, supervision, Writing – review and editing; Omer Dushek, Conceptualization, Funding acquisition, Investigation, Project administration, supervision, Writing – original draft, Writing – review and editing

### Author ORCIDs
Johannes Pettmann (iD) http://orcid.org/0000-0002-1979-8943
Anna Huhn (iD) http://orcid.org/0000-0003-4798-4951
Enas Abu Shah (iD) http://orcid.org/0000-0001-5033-8171
Mikhail A Kutuzov (iD) http://orcid.org/0000-0003-3386-4350
Michael L Dustin (iD) http://orcid.org/0000-0003-4983-6389
P Anton van der Merwe (iD) http://orcid.org/0000-0001-9902-6590
Omer Dushek (iD) http://orcid.org/0000-0001-5847-5226

### Ethics
Human subjects: Ethical approval was provided by the Medical Sciences Inter-divisional Research Ethics Committee (IDREC) at the University of Oxford (R51997/RE001).

### Decision letter and Author response
Decision letter https://doi.org/10.7554/eLife.67092.sa1
Author response https://doi.org/10.7554/eLife.67092.sa2

---

## Additional files

### Supplementary files
- Supplementary file 1. Sequences of 1G4 and A6 T cell receptors for lentiviral transduction.
- Supplementary file 2. Sequences of soluble A6 T cell receptors for surface plasmon resonance.
- Transparent reporting form

### Data availability
All data generated or analysed during this study are included in the manuscript and supporting files. Source data files have been provided for Figures 1, 2, 3, 4, 6, Figure 2—figure supplement 3, and Figure 4—figure supplement 1.

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

# Appendix 1

We provide information on each potency plot we generated in the sections that follow, including the location of the potency and $K_D$ data within each study. References to figures and tables are to those in the cited manuscript not to those in the present study. We provide the value of $\alpha$ produced and additional information, including $R^2$ and p-values are provided in *Figure 3—source data 1*, *Figure 6—source data 1*.

## Original mouse TCR data
### OT-I
### ID 1 (*Alam et al., 1996*)
Experiments were performed with the murine transgenic TCR OT-I that binds to a peptide from ovalbumin (OVA) presented on H2K$^b$. Affinity was measured by SPR at 25°C. Affinity values were taken from Table 1 and Figure 3f. For the power analysis, we used $K_D$ values estimated from the binding kinetics (kinetic $K_D$ values). Potency measures for the OVA peptide and peptide variants were previously measured by *Hogquist et al., 1995*. OT-I T cell responses to OVA and single amino acid peptide variants (A2 and E1) were measured in a cell lysis assay. For the power analysis, we extracted the potency of the peptides by reading the $P_{10}$ (peptide concentration producing 10% specific lysis) from dose-response curve in Figure 2. We excluded peptides that did not result in any response. We were able to include two data points with potency and affinity values for the power analysis producing $\alpha = 12$ (ID 1).

### ID 2 (*Alam et al., 1999*)
The OT-I TCR binding to the peptides derived from OVA was used with affinity and kinetics measured by SPR at 6, 25°C, and 37°C. Unusual biphasic binding was observed at 37°C for some peptides with two $k_{on}$ and two $k_{off}$ values reported based on a slow first and fast second step binding. Affinity values were provided in Table 1. To avoid picking the fast or slow phase parameters, we used the monophasic affinity data measured at 25°C for the power analysis. Potency data was taken from *Hogquist et al., 1995*. Three data points were included in the analysis producing $\alpha = 10$ (ID 2).

### ID 3 (*Rosette et al., 2001*)
OT-I TCR affinity and functional activity were measured when binding its wild-type ligand OVA or single amino acid variants (G4). Affinity values of TCR-pMHC interaction, measured by SPR at 25°C and 37°C, were provided in Table 1. Similar to *Alam et al., 1999*, TCR binding to MHC loaded with OVA showed biphasic binding at 37°C. As before, we used the data measured at 25°C for the power analysis. Functional data was generated with T cells isolated from OT-I transgenic mice. T cells were then stimulated with peptide-MHC complexes immobilised on plates. We read off potency data from dose-response curves in Figure 1. Only two data points were available for calculating the discrimination power $\alpha$ producing $\alpha = 18$ (ID 3).

### ID 4 (*Altan-Bonnet and Germain, 2005*)
In this study, the OT-I T cell response when stimulated with OVA was determined by phosphorylation of the kinase ERK in the MAPK pathway. Responses to OVA peptide as well as two peptide variants were studied. Potency values were extracted as $P_{10}$ from dose-response curves in Figure 1C. Only the OVA peptide could activate T cells above background. Potency for unresponsive peptides was set to the highest concentration used in assay. Therefore, the discrimination power $\alpha$ calculated with these data points gives a lower bound on the actual value for $\alpha$. Using the affinity data from *Alam et al., 1999* produced $\alpha > 5.1$ (ID 4).

### 3.L2
### ID 11 (*Kersh et al., 1998a*)
This paper contains affinity and kinetic data for 3.L2 TCR which recognises murine haemoglobin (Hb 64–76) measured by SPR at 25°C. We used $k_{off}$ values for the power analysis as $K_D$ values did not correlate (see main text). The D73 peptide was excluded from power analysis because this mutation impacted peptide loading to MHC. Potency data for this TCR was taken from *Kersh and Allen, 1996*. In the functional experiments, 3.L2 T cell hybridoma cells were incubated with antigen-presenting cells pulsed with peptides. Activation was measured by lysis of target cells.

$P_{40}$ values (ligand concentration at 40% lysis) of T cell response were given in Figure 4, and the corresponding dose-response curve was shown in Figure 5. Four data points were included in the analysis and produced $\alpha$ = 6.8 (ID 11).

### 2B4
### ID 14 (*Lyons et al., 1996*)
The 2B4 TCR used in this study recognises a moth cytochrome c (MCC) peptide bound to MHC class II molecule I-Ek. Table 2 provides $K_D$ values using SPR at 25°C. The potency of the peptides was determined with T cell hybridomas, stimulated by peptide-pulsed APCs, with activation determined by IL-2 production. For the power analysis, we extracted the $P_{10}$ from the dose-response curve in Figure 1A. Two data points were available for the power analysis producing an $\alpha$ = 6.7 (ID 14).

## Revised data for the original mouse TCRs
### OT-I
### ID 5 (*Stepanek et al., 2014*; *Daniels et al., 2006*)
Revised affinity data for OT-I TCR was published by *Stepanek et al., 2014*. The $K_D$ values were taken from the Table in Figure S1D. Potency data for the same set of peptide variants was measured by *Daniels et al., 2006*. Functional experiments were done with pre-selection of OT-I double-positive thymocytes. T cell activation was measured by expression of CD69 after incubation with peptide-pulsed antigen-presenting cells. $EC_{50}$ values, corrected for small differences in peptide affinity for MHC and normalised to OVA, were given in Figure 1a. Together, these papers provide five data points producing $\alpha$ = 2.1 (ID 5).

### ID 6 (*Stepanek et al., 2014*; *Zehn et al., 2009*)
*Zehn et al., 2009* provided additional functional data for OT-I TCR. Potency data is measured by intracellular IFNγ production by OT-I T cells stimulated with peptide pulsed antigen-presenting cell. The $EC_{50}$ values, given in table in Supplementary Figure 2C, were normalised to OVA. To calculate the discrimination power, we used $K_D$ values from *Stepanek et al., 2014*. The two data points available produced a power of $\alpha$ = 2.0 (ID 6).

### ID 7–10 (*Stepanek et al., 2014*; *Lo et al., 2019*)
*Lo et al., 2019* generated additional functional data for the OT-I TCR. Functional response of CD8+ or CD8- Jurkat cells expressing the OT-I TCR after stimulation with peptide pulsed antigen-presenting cells was measured by CD69 upregulation. The $EC_{50}$ values were provided in Supplementary Figure 7C. $K_D$ values were previously measured by *Stepanek et al., 2014*. The study included affinity and potency data for when one of the phosphorylation sites of LAT was mutated. The calculated discrimination power was the same ($\alpha$ = 1.1) for both wild-type LAT (ID 7) and mutated LAT (ID 9) unless CD8 was not present, in which case $\alpha$ = 0.37 (ID 8) or $\alpha$ = 1.4 (ID 10) using Jurkats expressing wild-type or mutated LAT, respectively.

### 3.L2
### ID 12 (*Persaud et al., 2010*)
In this study, the 3.L2 TCR as well as the M15 TCR, a high-affinity TCR engineered from the 3.L2 TCR system, were used, both TCRs bind to murine haemoglobin (Hb 64–76). Table 1 provides $K_D$ values using SPR with at 25°C. Functional data was generated by incubating T hybridoma cells with peptide-pulsed APCs and measuring IL-2 production. We extracted potency values from dose-response curves in Figure 1b and c. Potency values from both TCR systems produce $\alpha$ = 0.37 (ID 12).

### ID 13 (*Hong et al., 2015*)
This paper contains binding and potency data for the 3.L2 TCR interacting with the WT haemoglobin peptide and a panel of altered peptide ligands. Table 2 provides $K_D$ values using SPR at 25°C. The paper does not contain new potency measurement, and therefore, we used potency values measured by *Kersh and Allen, 1996* for the power analysis. This dataset produces $\alpha$ = 3.2 (ID 13).

## 2B4

### ID 17 (*Wu et al., 2002*)

This paper contains affinity data for 2B4 TCR binding to its cognate MCC antigen and a set of variant peptides. Table 1 contains $K_D$ values determined by SPR at 25°C. To compare with the original discrimination power, we used the original potency data (*Lyons et al., 1996*) to produce $\alpha$ = 2.8 (ID 17).

### ID 15–16 (*Krogsgaard et al., 2003*)

Revised data for 2B4 TCR is provided by *Krogsgaard et al., 2003*. Table 1 provides $K_D$ values measured by SPR at 25°C. For potency measurement, T cells from transgenic 2B4 mice were incubated peptide MHC molecules immobilised on plates, activation was measured by IL-2 production ($EC_{50}$ values were given in Table 1). All ligands were included in the analysis, including those initially labelled as outliers in the publication. The resulting α is 1.2 (ID 16). We also calculated α with affinity data from this study and potency data from *Lyons et al., 1996* ($\alpha$ = 2.2, ID 15).

### ID 18 (*Newell et al., 2011*)

*Newell et al., 2011* studied the 2B4 and the 226 TCRs that bind to MCC. The $K_D$ values were measured by SPR at 25°C and provided in Figures 5D (2B4) and 6B (266). T cell hybridomas were incubated with peptide-pulsed cells and T cell activation measured by IL-2 production. $P_{10}$ (concentration at 10% maximal IL-2 produced by wild-type 2B4) values given in Figure 5C (2B4) and 6 (266). Data for 2B4 produces $\alpha$ = 2.3 (ID 18). The 266 TCR was not included in the analysis because not enough data points were available.

### ID 19–20 (*Birnbaum et al., 2012*)

The affinity and potency of the 2B4 and the related 5cc7 TCR, which both interact with MCC, were reported. As before, SPR was used to report $K_D$ values. Functional assays were done with blasted transgenic T cell incubated with peptide pulsed cells. To determine potency, IL-2 production was measured. We extracted both $K_D$ values and $EC_{50}$ values from Figure 4C. Data for 2B4 produced $\alpha$ = 0.95 (ID 19) and for 5cc7 produced $\alpha$ = 0.74 (ID 20).

## Other mouse TCRs

### P14

### ID 21–22 (*Tian et al., 2007*)

The mouse P14 TCR that recognises a set of altered peptides from the lymphocytic choriomeningitis virus epitope gp3341 on murine class I MHC Db. All binding parameters were measured by SPR at 25°C. In functional assays, T cell cytotoxicity and IFN-γ production of blasted splenocytes from P14 TCR transgenic mice were measured when binding peptide-MHC. Cytotoxicity was measured in a cellular assay, IFNγ production in a plate assay. The $EC_{50}$ is used as potency measurement. All affinity and potency data were provided in Table 2. The α values for this TCR system are 2.1 for cytotoxicity assay (ID 21) and 1.3 for IFNγ assay (ID 22).

### B3K506 and B3K508

### ID 23–26 (*Govern et al., 2010*)

The MHC-II restricted B3K506 and B3K508 TCRs that recognise the 3K peptide were studied. The $K_D$ values were measured by SPR at 25°C. T cell response was measured by T cell proliferation and cytokine production after stimulation with peptide-pulsed APCs. All $K_D$ and $EC_{50}$ values were given in Table S1. The B3K506 system produced $\alpha$ = 2.9 (ID 23) and $\alpha$ = 2.4 (ID 24) and the B3K508 system produced $\alpha$ = 2.9 (ID 25) and $\alpha$ = 2.5 (ID 26) for proliferation and TNFα production, respectively.

### 2C

### ID 27–29 (*Chervin et al., 2009*)

A panel of TCRs, derived from the murine 2C TCR, that differed in their affinity to the SIYR peptide presented on H-2K$^b$ was used. The $K_D$ values were measured by SPR at 25°C and provided in Table 1. Functional experiments were done with T cell hybridomas with or without CD8 expression. T cells were either incubated with peptides immobilised on plates or with antigen-presenting cells

pulsed with peptides. For cellular experiments, $EC_{50}$ values are given in Figure 3B and D (with and without CD8, respectively), for plate assays in Figure 4 (only CD8 negative data). Most of the ligands have a $K_D < 1$ µM, hence the data points were excluded from the analysis (see inclusion/exclusion criteria in Materials and methods), and therefore, only few data points remained for the power analysis. CD8-negative T cell expressing TCRs stimulated in a plate assay produced $\alpha = 0.12$ (ID 27); however in the cellular assay; TCRs binding to the antigen with a $K_D < 1$ µM were not activated in CD8-negative T cells (no data points to calculate α) (ID 28). TCRs in CD8-positive T cells stimulated in the cellular assay produce $\alpha = 0.66$ (ID 29).

### ID 30 (*Bowerman et al., 2009*)
The 2C high-affinity TCR and variants thereof binding to the QL9 and the altered QL9 peptide F5R were studied. The $K_D$ values were measured by SPR at 25°C. Functional data was generated with T cell hybridomas stimulated by peptide-pulsed APCs with T cell activation assessed by IL-2 production. $K_D$ and $EC_{50}$ values were taken from Table 1. $K_D$ values below 1 µM were excluded from our power analysis. This data produces $\alpha = 2.7$ (ID 30).

### ID 31–32 (*Jones et al., 2008*)
The authors report binding and functional responses of high-affinity 2C TCR variants interacting with SIY peptide on MHC $K^b$ and QL9 peptide on $L^d$. In total, eight different TCR/pMHC ligand pairs were included. The $K_D$ values were measured by SPR at 25°C and provided in Table 1. $K_D$ values lower than 1 µM were excluded from the analysis. Functional assays were done with T cell hybridoma with and without CD8 expression with T cell activation assessed by IL-2 production in response to peptide-pulsed APCs. We extracted potency values as $P_{50}$ from dose-response curves in Figure 3. TCR variants m6 and m13 when binding to SIY-$K^b$ showed no activation ($P_{50} > 100$ µM). The calculated discrimination power is $\alpha = 4.7$ for CD8-positive (ID 31) and $\alpha = 6.5$ for CD8-negative T cells (ID 32).

### Not included (*Holler and Kranz, 2003*)
This study provided binding and affinity data for the 2C TCR with and without CD8. However, when applying our inclusion/exclusion criteria only a single data point was available, and therefore, we were unable to calculate α. The reason is that only a few interactions were measured by SPR and the majority of these produced $K_D$ values below 1 µM.

## 42F3
### ID 33 (*Adams et al., 2016*)
The 42F3 TCR recognises the class I MHC molecule H2-Ld presenting the peptide p2Ca. The $K_D$ values were measured by SPR at 25°C and potency data ($EC_{50}$ of IL2 production after cellular stimulation) were taken from Table 1 and Supplementary Figure 3C. The resulting α is 0.15 (ID 33).

## Gp70 (AH1)-specific TCR
### ID 34 (*McMahan et al., 2006*)
The TCR used in this study recognises the AH1 peptide which is derived from the endogenous retroviral protein gp70(423–431), a MHC class I restricted tumour-associated antigen. The authors used a set of AH1 variants with optimised affinities. The $K_D$ values were measured by SPR at 25°C and provided in Figure 1B. Functional data was generated with a T cell line incubated with peptide-pulsed APCs. $EC_{50}$ values of a proliferation assay are provided in Figure 2B. The calculated discrimination power was $\alpha = 5.2$ (ID 34).

## Other human TCRs
### 1G4
### ID 35 (*Irving et al., 2012*)
The 1G4 TCR used in this study binds the NY-ESO-1 (157–165) peptide loaded on MHC class I HLA-A2. The authors generated a panel of TCRs derived from the human 1G4 TCR that bind with higher affinity than the wild-type TCR. The $K_D$ values were measured by SPR at 25°C and provided in a table in Figure 1A. Potency was measured with a cytotoxicity assay, and we extracted the mean $EC_{50}$ values from Figure 5E . A decrease in potency was observed for TCRs with an affinity

of $K_D < 1$ μM, which were excluded as per our exclusion criteria (see Materials and methods). This data produced $\alpha = 0.67$ (ID 35).

### ID 36 (*Da et al., 2010*)

Here, TCR peptide MHC binding parameters and T cell function were investigated with a panel of 1G4 TCR variants binding to the NY-ESO-1 peptide. The $K_D$ values were measured by SPR and provided in Table 1. The functional response of T cells was determined in a cytotoxic T cell assay. We extracted the mean $EC_{50}$ values from Figure 4B. Data points with $K_D < 1$ μM are excluded from the power analysis. The resulting α is 0.69 (ID 36).

### ID 37–38 (*Aleksic et al., 2010*)

Here, the interaction between 1G4 TCR binding a set of variant NY-ESO-1 (157–165) peptides on MHC class I was studied. The $K_D$ values were measured by SPR at 37°C. The potency was determined by IFNγ production of T cell after stimulation by plate-immobilised pMHC or cytotoxicity by peptide-pulsed T2 APCs. The 1G4 TCR clone was used for both experiments. All affinity and $EC_{50}$ values were given in Table S1. Discrimination power α for the 1G4 system is 0.6 (IFNγ, ID 37) and 1.6 (cytotoxicity assay, ID 38).

## 1G4 and G10

### ID 39–40 (*Dushek et al., 2011*)

Experimental data was generated with the 1G4 and G10 TCR clones binding to a panel of peptide variants. The 1G4 TCR recognises the NY-ESO-1 antigen, and the G10 TCR recognises the HIV gag p17 antigen in the context of MHC class I HLA-A2. The $K_D$ values were measured by SPR at 37°C. Potency was determined by measuring IFNγ production in response to plate-immobilised recombinant pMHC. All $K_D$ and $EC_{50}$ values were given in Table S1 and S2. For the 1G4 system, we found $\alpha = 0.55$ (ID 39); and for the G10 system, we found $\alpha = 0.95$ (ID 40).

## 1E6

### ID 41–42 (*Cole et al., 2016*)

The MHC-I-restricted 1E6 TCR reactive to preproinsulin (INS) and variants were studied. The $K_D$ values were measured by SPR at 25°C and 37°C and provided in Figure 2. All $K_D$ values lower than 1 μM were excluded from the power analysis (see Materials and methods). Functional assays were done with primary T cells responding to peptide-pulsed APCs and target cell lysis was measured for T cell activation. The $EC_{50}$ was determined from the data in Figure 2K. We calculated $\alpha = 1.1$ for $K_D$ values measured at 25°C (ID 41) and $\alpha = 1.2$ for $K_D$ values measured at 37°C (ID 42).

## A6

### ID 43–44 (*Thomas et al., 2011b*)

The A6 and engineered variants recognising the Tax or HuD peptides were used. The $K_D$ values were measured by SPR at 25°C and provided in Figure 1A. T cell activation in response to peptide-pulsed APCs was assessed by CD107a expression and IFNγ production. Potency data was extracted as $P_{20}$ for CD107a assay from dose-response curve in Figure 4C and as $P_{10}$ for IFNγ assay from dose-response curve in Figure 5A. Data point with $K_D < 1$ μM was not included in the power analysis. The resulting α is 2.0 (ID 43) and 2.2 (ID 44) for CD107 and IFNγ readout, respectively.

### Gp100-specific TCR (melanoma)

### ID 45–46 (*Zhong et al., 2013*)

Seven TCRs specific to human melanoma gp2092M epitope (modified from gp100 (209–217)) were isolated from patients vaccinated with gp2092M. The $K_D$ values of these TCRs measured by SPR at 25°C were provided in Table 1. Functional activity was determined by IFNγ production and ERK phosphorylation of transduced CD8+ splenocytes mixed with peptide-pulsed APCs. Potency values were extracted from Figure S3A and C as $P_{10}$. The L2G2 TCR, which appeared as an extreme outlier showing the highest potency despite having the lowest affinity, was excluded from the analysis. This data point is shown in the plots as an open circle and including it would have further reduced the estimates α. The calculated powers were $\alpha = 1.3$ for IFNγ production (ID 45) and $\alpha = 1.2$ for ERK phosphorylation assay (ID 46).

### ID 47–48 (*Bianchi et al., 2016*)

T cell responses of a TCR specific to melanoma epitope gp100(280–288) were studied using a set of altered peptides. The $K_D$ values were measured by SPR at 25°C and provided in Table 2. Functional assays used gp100 TCR-transduced CD8+ T cells stimulated by peptide-pulsed APCs with T cell activation assessed by cytotoxic lysis and MIP-1β production. We extracted the potency data as $P_{10}$ from dose-response curves in Figure 6. The resulting α values were 2.3 (lysis assay, ID 47) and 3.6 (MIP-1β production, ID 48).

## 14.3.d

### ID 49–50 (*Andersen et al., 2001*)

T cell responses were measured using variants of the *Staphylococcus* enterotoxin C3 (SEC3) super antigen. In addition, binding of a panel of mutated variants of the antibody F23.1 was also used. The $K_D$ values of SEC3 were measured by SPR and provided in Table 1. The $K_D$ values of the antibodies were provided in Table 1 of different publication (*Andersen et al., 2001*). T cell hybridomas, containing a NFAT-GFP expression cassette, were stimulated with SEC3 or antibody molecules immobilised onto plate surfaces to observe functional responses. We extracted all potency values as EC20 from Figure 4. According to our exclusion criteria (see Materials and methods), we did not include any data point where $K_D < 1$ μM. The remaining data points generated with the SEC3 variants produced $\alpha = 0.81$ (ID 49) and with the F23.1 antibody variants produced $\alpha = 0.66$ (ID50).

## TCR55

### ID 51 (*Sibener et al., 2018*)

This study used TCRs specific for HLA-B35-HIV(Pol448–456) binding to a set of variant peptides. The $K_D$ values were measured by SPR at 25°C. T cell activation after stimulation with peptide pulsed on APCs was measured by CD69 expression. All $K_D$ and $EC_{50}$ values were given in Figure S5C. We calculated $\alpha = 0.19$ (ID 51).

## ILA1

### ID 52–57 (*Laugel et al., 2007*)

The MHC class I-restricted ILA1 TCR is specific for the human telomerase reverse transcriptase (hTERT) epitope ILAKFLHWL (hTERT540-548). The $K_D$ values were measured by SPR at 25°C and provided in Table 1. Three different assays were used to measure T cell activation: degranulation assay, CD107a expression, and IFNγ production. Each assay was performed using APCs expressing either WT MHC or CD8-null MHC which cannot bind CD8. Potency values for degranulation were given in Table 1, CD107a and IFNγ potency data was extracted from dose-response curves in Figure 7. For potency data measured with wild-type (WT) and CD8 null MHC, respectively, we calculated an α of 1.5 (WT, ID 52) and 2.5 (CD8 neg., ID 52) for degranulation, 2.2 (WT, ID 54) and 3.6 (CD8 neg., ID 55) for CD107a, and 2.2 (WT, ID 56) and 3.2 (CD8 neg., ID 57) for IFNγ production.

### ID 58–61 (*Tan et al., 2015*)

The ILA1 TCR was studied interacting with peptide variants. The $K_D$ values were measured by SPR at 25°C and provided in Table 1. T cell activation was measured by peptide-pulsed APCs and determined by MIP-1β, IFNγ, TNFα, and IL-2 production using intracellular cytokine staining. The potency values were read of as $P_{50}$ from the dose-response curves in Figure 2. Authors suggested that the TCR shows a plateau at $K_D$ values < 5 μM. Therefore, we decided to exclude $K_D$ values < 5 μM from the power analysis to avoid underestimating the discrimination power α. The data produces $\alpha = 1.4$ (ID 58), 0.77 (ID 59), 0.97 (ID 60), and 1.1 (ID 61) for MIP-1β, IFNγ, TNFα, and IL-2 production, respectively.

## NY-ESO-1 (60–72)-specific TCR

### ID 62 (*Chan et al., 2018*)

Four TCRs binding to the tumorigenic antigen NY-ESO-1 (60–72) were obtained from patients with melanomas expressing NY-ESO-1. The $K_D$ values were measured by SPR 25°C and given in Figure 2C. Functional response of TCRs to exogenous peptide stimulation was assessed by measuring IFNγ production of T cells incubated with NY-ESO-1-expressing melanoma cells. We extracted $EC_{50}$ values from dose-response curves in Figure 1F. We calculated $\alpha = -0.59$ (ID 62).

### Gliadin-specific TCRs (celiac disease)
### ID 63 (*Broughton et al., 2012*)
Seven DQ8-glia-a1-restricted TCRs isolated from celiac disease patients were characterised for their binding affinity to a-I- gliadin and their functional response. The $K_D$ values were measured by SPR at 25°C and provided in Figure 2 and S5. T cell activation was assessed by proliferation in response to peptide-pulsed APCs. We extracted $P_{20}$ values from the dose-response curves in Figure 1 (black curve Q-Q). We calculated an $\alpha = 0.83$ (ID 63).

### LC13
### ID 64–67 (*Burrows et al., 2010*)
The LC13 and SB27 TCRs were studied using an alanine scan. The $K_D$ values were measured by SPR and provided in Table S2. T cell activation was measured using Jurkat T cells expressing the TCR with CD69 and cytotoxicity assessed in response to peptide-pulsed APCs. Figure 1C and 1D showed the dose-response curves for CD69 upregulation for either CD8-positive or CD8-negative cells. We extracted the $P_{30}$ as potency measure. $EC_{50}$ of cytotoxicity assay was given in Figure 2 for LC13 and Figure S2 for SB27. Potency values from CD69 produced $\alpha = 1.9$ (ID 64) for CD8-positive cells and $\alpha = 7.8$ (ID 65) for CD8-negative cells. Lysis assays produced $\alpha = 4.1$ (ID 66) for the LC13 and $\alpha = 0.11$ (ID 67) for SB27 TCR.

### HIV-Gag293-specific TCRs
### ID 68 (*Benati et al., 2016*)
TCRs specific to HIV Gag293 protein were isolated from patients infected with HIV. The $K_D$ values were measured by SPR and provided in Table 3. T cell activation was assessed using TCR-transduced J76 cells measuring CD69 expression in response to peptide-pulsed APCs. We extracted the mean $EC_{50}$ values from Figure 6D. We calculated $\alpha = 1.0$ (ID 68).

### MEL5
### ID 69 (*Ekeruche-Makinde et al., 2012*)
The MEL5 and MEL187.c5 TCRs were studied that bind the MART-1 antigen and variants thereof. The $K_D$ values were measured by SPR at 25°C and provided in Table 1. T cell activation was measured by MIP-1β production in response to peptide-pulsed APCs. We extracted potency values as $P_{50}$ from dose-response curves in Figure 2 and S1. Because responses to peptides were measured in separate experiments, potency data is normalised to wild-type peptide. This produced $\alpha = 2.3$ (ID 69).

### ID 70 (*Madura et al., 2019*)
The MEL5, MEL187.c5, DMF4, and DMF5 were studied that recognise the MART-1 antigen. Two overlapping peptides were used: nonapeptide MART-1 (27–35) and decapeptide MART-1 (26–35). The $K_D$ values were measured by SPR at 25°C and provided in Table 1. T cell activation was assessed using primary human T cells responding to peptide-pulsed APCs with MIP-1β used as a marker of T cell activation. We determined $P_{30}$ directly from does-response curves in Figure 1A. Data produced $\alpha = 4.5$ (ID 70).

## Other (non-TCR) surface receptors
### Cytokine receptors
### ID 1–2 (*Levin et al., 2012*)
Engineered IL-2 variants with increased binding affinity for theIL-2 receptor subunit β (IL-2Rb) were studied. The $K_D$ values for IL-2 variants to IL-2Rβ are given in Supplementary Figure 3 and determined by SPR at 25°C. As only the affinities to a single subunit were varied between ligands, potency was plotted over these $K_D$ values. Functional experiments were performed with either CD25-negative human Natural Killer cells or CD25-negative murine T cells. We extracted the $EC_{50}$ values as a measure of potency from dose-response curves in Figure 3a and 3e. We calculated $\alpha = 0.55$ (ID 1) for experiments done with Natural Killer cells and $\alpha = 0.74$ (ID 2) for T cells.

### ID 3–6 (*Moraga et al., 2015*)

The relationship between the interaction of IL-13 with its cytokine receptor and the resulting downstream cellular responses was investigated. A panel of IL-13 variants with a range of binding affinities for the receptor subunit IL-13Ra1 was generated. Binding affinities of these ligands were given in a table in Figure 2C. Here, only the affinity for the α subunit of the receptor dimer was varied, and therefore, we plotted potency over these $K_D$ values. Functional responses of binding were determined by measuring STAT6 phosphorylation, CD86 and CD209 production, and proliferation after receptor stimulation. We extracted $EC_{50}$ values for pSTAT6 from Figure 5B. To avoid extrapolating potencies, ligands with $EC_{50}$ larger than highest concentration used in the dose-response (in Figure 5A) were excluded. The mean proliferation $EC_{50}$ values were taken from Figure 5G. CD86 $EC_{50}$ values were extracted from dose-response curve in Figure 5H, CD209 $EC_{50}$ values from the dose-response curve in Figure S7C. $EC_{50}$ values for CD86 and CD209 extracted from the dose-response curves did not exactly match $EC_{50}$ values given in Figure S7 D and E, but both values resulted in similar α values. The α values calculated for the IL-13 receptor are 0.47 (ID 3), 0.39 (ID 4), 0.44 (ID 5), and 0.42 (ID 6) for potency values from pSTAT6, proliferation, CD86, and CD205 assays, respectively.

### ID 7–8 (*Thomas et al., 2011a*)

This study uses a set of mutated cytokines derived from IFNα2 and IFN$\omega$, binding cytokine receptors IFNAR1 and IFNAR2. All binding affinities of mutants normalised to WT are provided in Supplemental Table 2. Because mutations change the affinities to both IFNAR1 and IFNAR2, we calculated an effective binding affinity by multiplying $K_D$ of IFNAR1 with $K_D$ of IFNAR2 (R1xR2). Functional response of cells to cytokine mutants was determined by their antiviral activity in a Hepatitis C Virus Replication Assay, their antiproliferation activity on WISH cells. Mean $EC_{50}$ values normalised to WT obtained from Figure 7A. We calculated $\alpha = 0.71$ (ID 7) for antiviral potency and $\alpha = 1.3$ (ID 8) for antiproliferation potency.

### ID 9–11 (*Mendoza et al., 2017*)

Study of IFN1 receptor activation with engineered higher-affinity type I IFNs. Affinity constants for peptides to each receptor subunit were measured by SPR. To get the effective $K_D$, we multiplied $K_D$ of IFN-αR1 binding with $K_D$ of IFN-αR2 binding (R1xR2) Ligand activity was measured by STAT phosphorylation, antiviral activity, and antiproliferation activity. All affinity and $EC_{50}$ values were provided in Table S2. The data produced $\alpha = 0.024$ for STAT1 phosphorylation (ID 9), $\alpha = 0.034$ for antiviral activity (ID 10), and $\alpha = 0.50$ (ID 11) for the anti-proliferation assay.

### ID 12–13 (*Martinez-Fabregas et al., 2019*)

In this study, the authors engineered IL-6 variants with different affinities to the IL-6 receptor subunit gp130. Cytokine gp130 binding kinetics were measured with a switchSENSE chip, binding parameters were given in Supplementary Figure 1D. The influence of IL-6 variants on functional activity of the receptor was determined by the amount of STAT1 and STAT3 phosphorylation at different ligand concentrations. We read off the potency of each ligand as $P_{25}$ directly from dose-response curves in Figure 2A and B. We calculated $\alpha = 0.54$ for pSTAT1 (ID 12) and $\alpha = 0.52$ for pSTAT 3 (ID 13).

## Receptor tyrosine kinase (RTK)

### ID 14 (*Reddy et al., 1996*)

In this study, the effect of three mutated epidermal growth factor on epidermal growth factor receptor (EGFR) was studied. Affinity values of growth factor to receptor were measured with radioactive labelled ligands binding to receptors on cells. Data are given in Table 1. Functional response of cells to ligands was determined by measuring the specific growth rate after stimulation. We extracted the $EC_{50}$ values from dose-response curves in Figure 4. This produced $\alpha = 0.55$ (ID 14).

### ID 15–16 (*Ho et al., 2017*)

Paper contains data on the c-Kit receptor tyrosine kinase which is activated by the SCF. Affinity and functional response of the receptor to SCF variants was studied. Binding parameters were measured by SPR and provided in Figure 1F. Cell activation after stimulation with ligands was determined by the amount of ERK and AKT phosphorylation (pERK and pAKT). We extracted

the potency data for each variant as $EC_{50}$ from dose-response curves in Figure 2D and 2E. We calculated $\alpha$ = 0.83 (ID 15) and $\alpha$ = 0.88 (ID 16) for pERK and pAKT measurements, respectively.

## GPCRs
### ID 17–18 (*Guo et al., 2012*)
The binding parameters of the GPCR adenosine A2A receptor to various agonist and their functional effects were studied. Association and dissociation rates, and hence $K_D$ values, were determined with a kinetic radioligand binding assay. Functional activity of HEK293 expressing the A2A receptor was measured by detecting cAMP production and changes in cell morphology. The binding data was provided in Table 3, and $EC_{50}$ values from functional experiments were given in Table 4. The discrimination power calculated with cell morphology data is $\alpha$ = 0.29 (ID 17) and with the cAMP assay produced $\alpha$ = 0.71 (ID 18).

### ID 19–20 (*Sykes et al., 2009*)
The M3 muscarinic receptor was studied using a set of agonist. The binding kinetics were determined with competition binding assay and were provided in Table 1. Agonist potency was measured by guanosine 5'-O- (3-[35 S]thio) triphosphate (GTPγS) binding to GαD subunits, and by intracellular calcium levels after receptor stimulation. Potency data measured as $EC_{50}$ values were provided in Table 2. The resulting power was $\alpha$ = 0.77 (ID 19) and $\alpha$ = 0.55 (ID 20) for calcium response and GTPγS binding assay, respectively.

### ID 21 (*Guyon et al., 2013*)
The CXCR4 receptor is activated by the chemokine CXCL12. In this paper, the interaction of Baclofen and other GABA ligands was tested on their abilities to activate CXCR4. The affinity of ligands to the receptor was measured by back-scattering interferometry, and $K_D$ values given in Figure 7. Functional response of oocytes expressing CXCR4 to ligands was determined by measuring the inward currents at different ligand concentrations. $EC_{50}$ values were provided in Table 1. We calculated $\alpha$ = 0.57 for this system (ID 21).

### ID 22–24 (*Heise et al., 2005*)
Characterisation of binding properties and potencies of CXC chemokine receptor 3 antagonists. Binding properties of antagonist were determined using kinetic radioligand binding assay. Affinity values were in Table 1 measured for different cell lines. Functional responses after ligand binding were measured guanosine 5'-O-(3-[35 S]thio)triphosphate (GTPyS) binding, calcium release, and cellular chemotaxis. All $EC_{50}$ values of assays were given in the text. We calculated $\alpha$ = 0.72 (ID 22), 1.1 (ID 23), and 0.56 (ID 24) for calcium release, GTPyS binding, chemotaxis assays, respectively.

## CARs
### ID 25 (*Chmielewski et al., 2004*)
This study contains affinity and potency data for a CAR binding the ErbB2 surface antigen. The authors generated a series of anti-ErbB2 single-chain variable fragments fused to the CD3 $\zeta$ cytoplasmic domain. The $K_D$ values are reported in Table 1. Functional experiments were done in a plate assay, with ErbB2 immobilised to a surface. Potency of receptors was measured by IFNγ production of T cells after stimulation. We extracted $P_{20}$ values from dose-response curve in Figure 4A. The resulting $\alpha$ is 0.52 (ID 25).

### ID 26–27 (*Liu et al., 2015*)
This study characterised a panel of CARs that bind to the ErbB2 surface protein. CARs were constructed by linking the various anti-ErbB2 single-chain variable fragments to the CD8αD hinge and transmembrane domain followed by the 4-1BB and CD3 $\zeta$ intracellular signalling domains. The $K_D$ values were measured by SPR and provided in Table S1. For functional experiments, CAR T cells were incubated with ErbB2-expressing cells. We obtained potency data by using CD107a expression and proliferation assay data in Figure 2A and C to the respective plot dose-response curves. $P_{50}$ values were extracted from these plots. The resulting $\alpha$ values are 1.1 for CD107 (ID 26) and 0.64 for proliferation assay (ID 27).

## ID 28 (*Taylor et al., 2017*)

*Taylor et al., 2017* developed a synthetic CAR signalling system in which the extracellular domains of the CAR and its ligand antigen were exchanged with short hybridising strands of DNA. The DNA-CAR $\zeta$ consists of a ssDNA covalently attached to a SNAP tag protein which was fused to a transmembrane domain and the CD3 $\zeta$ chain. Stands of different length and sequence were designed to vary the affinity of the CAR to the ligand. Binding was measured as the lifetime ($\tau_{corr}$) of single ligand-CAR interactions using microscopy and corrected for photobleaching and provided in Figure 2D. The dissociation rate $k_{off}$ was calculated from the lifetimes with $k_{off} = ln(2)/\tau$. To measure T cell responses, ligands, consisting of the complimentary strand of ssDNA, were anchored in planar-supported lipid bilayer where they can freely diffuse. The DNA-CAR $\zeta$ was expressed in TCR-negative Jurkat cells. Cell activation after incubation with ligands was measured by phosphorylation of ERK. Potency data was extracted as $P_{20}$ from dose-response curves in Figure 2C. This CAR system produced $\alpha = 1.2$ (ID 28).

## BCRs

### ID 29–30 (*Batista and Neuberger, 1998*)

The study used the HyHEL10 and D1.3 BCRs, which have a high affinity to the hen egg lysozyme (HEL) and variants thereof. The $K_D$ values were measured by SPR at 25°C, and dissociation rates were provided in Table 1. For functional experiments, the ability of B cells to mediate HEL presentation to T cell hybridomas after stimulation with mutant lysozymes was determined by measuring IL-2 production of T cells specific to HEL. We extracted the potency data from doseresponse curves in Figures 3 and 4 as $EC_{50}$. The authors described an affinity floor for the B cell receptor when the dissociation rate was below $10^{-4}$ s-1 so that potency did not longer decrease for these interactions. To avoid underestimating $\alpha$, we did not include these higher-affinity ligands in the power analysis. The resulting $\alpha$ values were $\alpha = 1.4$ for the D1.3 BCR (ID 29) and $\alpha = 1.3$ for the HyHEL10 BCR (ID 30).

