## [Decision Letter]

**Acceptance summary:**

The presented manuscript takes a very comprehensive and elaborated look at how T cell receptors (TCR) discriminate between self and non-self antigens. By combining experimental and analytical methods, the presented findings challenge commonly held notions and could be fundamental for our understanding of the T cell immune response, with implications for autoimmunity and immunity to cancer.

**Decision letter after peer review:**

Thank you for submitting your article "The discriminatory power of the T cell receptor" for consideration by *eLife*. Your article has been reviewed by 3 peer reviewers, one of whom is a member of our Board of Reviewing Editors, and the evaluation has been overseen by Tadatsugu Taniguchi as the Senior Editor. The reviewers have opted to remain anonymous.

Essential revisions:

All reviewers appreciated the elaborated and thorough analyses presented in this paper. The detailed comments (see below) address some specific points that the reviewers noted. In particular this concerns the following main aspects that should be addressed in a revised version of this manuscript:

1. Please address some points regarding possible overfitting and parameter identification within the mathematical analyses.

2. Please comment on the ability of the current Kinetic proofreading model to explain antagonism

The detailed comments of the reviewers are as follows:

*Reviewer #1 (Recommendations for the authors):*

– The authors state that when fitting the 2-parameter Hill-function to the TCR binding curves, that different combinations of B_max_ and K_D_ are suggested to explain the data if B_max_ was estimated. The coefficient of variance was used to assess this effect but wouldn't it be more straightforward to use methods such as Profile likelihood analysis to clearly address parameter non-identifiability?

*Reviewer #2:*

1. Please further clarify and explain the extrapolation of the connection between B_max_ of the TRC-peptide binding curve and W6/32 binding curves based on Figure 1D.

1.1. Cosmetic – the color coding in 1B is a bit confusing because the same color palette denotes different concentrations of the same peptide here as the one denoting different peptides in 1D.

2. Please address the potential overfitting issue in the KPR model, and the parameter value spread in Figure S8.

3. Please comment on how the current model relates to antagonism, which is commonly considered a typical feature of TCR signaling

*Reviewer #3:*

As came probably through already in the other review sections I feel enthusiastic about the work presented as it combines a massive volume of bench work-related data with relatively simple math (with the exception of the section dealing with KP) and it leads to testable clear hypotheses. The latter may be the tasks of others (or not), and hence I consider the manuscript close to being appropriate for publishing.

Here are a few suggestions I consider useful for the reproducibility and readability by others.

i. A detailed description of the SPR-protocol including primary data would render the reading more complete. The methods section is already fairly detailed, but it would be more than helpful to understand how the streptavidin and pMHC-bio run, as well as the TCR-runs and the final W6/32 run were computed against one another to arrive a the B_max_ of low affinity ligands. TCR-binding to pMHC (even if saturated) will result in a different SPR signal than antibody binding (given the different molecular masses). Since this methodology is central to the manuscript, I would argue in favor of introducing it more thoroughly in Figure 1A (and B, with representative raw data leading to corrected B_max_). This may also help to understand why the addition of W6/32 in both panels of figure 1B led to a much lower signal than the highest concentration of TCR.

ii. After lentiviral infection of primary T cells, how did the authors monitor the expression levels of the introduced TCRs? How did they verify the absence of mixed TCR dimers? Since TCR expression levels are clearly important in the overall scenario analyzed, I believe this is a fair question to ask.

iii. I find it somewhat unusual (if not irritating) to find already published data in main figures. An in my opinion better approach is to cite the work and show exclusively the processed data (taken from cited work, in the lower panel of Figure 3A, B,C etc.).

iv. The Discussion section would benefit from a short paragraph reflecting on work on TCR-pMHC binding as measured within the confines of the immunological synapse. Reasons are that binding constants (if there is such a thing in synapses) differ in some cases substantially from what has been measured via SPR for reasons that are still being debated. Clearly, one major strength of this manuscript is that cellular behavior correlates with SPR measurements of TCR-pMHC binding. Nonetheless scenarios, especially those involving KP, may turn out differently in case serial TCR engagement by few antigens is supported.

[Editors' note: further revisions were suggested prior to acceptance, as described below.]

Thank you for resubmitting your work entitled "The discriminatory power of the T cell receptor" for further consideration by *eLife*. Your revised article has been evaluated by Tadatsugu Taniguchi as the Senior Editor and a Reviewing Editor, as well as the original reviewers of your paper.

They all appreciate the elaborated and thorough analyses presented in the paper, and also appreciate the additional work that had been done to address the previous comments of the reviewers. The manuscript has been improved, but there are some remaining issues that need to be addressed, as outlined below:

1. One remaining concern mainly addresses the interpretation of the relationship between the values of B_max_ and the AB binding to pMHC in Figure 1D. The data suggest that this linear relationship could be extrapolated to very weak pMHC's but it does not need to be the case (see also comment reviewer 2). Although this is acknowledged in your statement (line 95-96), we recommend to discuss this in more detail. It seems that this assumption mainly drives the selection for the method to analyze the low affinities, with B_max_ constrained by this relationship. We would recommend to acknowledge this in your discussion or provide additional evidence.

2. There are some issues with regard to the presentation in Figure 1D that should be addressed.

Please also see the specific comments of the reviewers below.

*Reviewer #2 (Recommendations for the authors):*

I am still somewhat confused about the correspondence between the TCR B_max_ of binding to pMHCs in figure 1 B, but I think the importance of the results is compelling, and I am happy if this debate plays out in the literature rather than here. To summarize:

1. It still looks (to me) that the linear relationship between Ab binding to pMHC and the B-max of the TCR binding is just an empirical finding, and although the data does suggest it might continue for very weak pMHC's I don’t see why it would be guaranteed. On the same note, the authors say in their response, " W6/32 is a conformationally-sensitive antibody that does not depend on the precise peptide sequence" – which on its face contradicts the fact that the response to W6/32 is different to different peptides in figure 1 D.

Please discuss.

2. I am still confused about the legend in Figure 1D. For instance, black dot is supposed to be Tax WT pMHC for A6 TCR – why are there four different black dots in the plot, with completely different B-max? Same question for NYE 9V. Finally filled orange dots are supposed to be Tax 1M. So what are the empty orange circles – they are not indicated in the legend, as far as I can see? The open purple circles appear in the plot but not in the legend. By contrast, open pink circle is indicated in the legend but does not seem to appear in the plot. Etc. If this is just mis-labeling it should be corrected and better explained in the text/caption.

3. Figure 1C, left is supposed to show the binding curve for WT NY-ESO-1 pMHC but it is not shown in Figure 1B – why? Similarly, I could not find in the text what peptide is the Figure 1C, right for? In either case, it should be better explained in the text/caption

*Reviewer #3 (Recommendations for the authors):*

The authors have addressed all issues I have raised in my previous review in a satisfactory manner. I enthusiastically recommend publishing their work in *eLife*.

---

## [Author Response]

Essential revisions:All reviewers appreciated the elaborated and thorough analyses presented in this paper. The detailed comments (see below) address some specific points that the reviewers noted. In particular this concerns the following main aspects that should be addressed in a revised version of this manuscript:1. Please address some points regarding possible overfitting and parameter identification within the mathematical analyses.

We have now increased the amount of information we provide when describing data fitting. In the case of fitting the SPR data to determine K_D_, we have revised the Results section and the methods section. In relation to fitting the kinetic proofreading model, we have explicitly stated the number of fitted parameters and the number of data points, which supports our conclusions based on ABC-SMC that we can uniquely identify the KP parameters.

2. Please comment on the ability of the current Kinetic proofreading model to explain antagonism

We have now added a discussion paragraph to explain that the basic proofreading model we have used cannot explain antagonism. This comment prompted us to also consider the fact that the basic proofreading model cannot explain another feature of T cell responses, namely the observation of an optimal pMHC affinity for T cell activation. Overall, we have aimed to explain that our model can explain antigen discrimination in typical scenarios where it is presented on autologous cells in the presence of self pMHC but additional data, and modifications to the basic kinetic proofreading model, are needed to explain antagonism and optimal pMHC affinity..

The detailed comments of the reviewers are as follows:Reviewer #1 (Recommendations for the authors):– The authors state that when fitting the 2-parameter Hill-function to the TCR binding curves, that different combinations of B_max_ and K_D_ are suggested to explain the data if B_max_ was estimated. The coefficient of variance was used to assess this effect but wouldn't it be more straightforward to use methods such as Profile likelihood analysis to clearly address parameter non-identifiability?

Thank you for this suggestion that we have now investigated. We produced the profiles for the high affinity (where we expect to identify both K_D_ and B_max_) and the low affinity (where we do not expect to identify these parameters) pMHC data from Fig 1C in the main text. As the reviewer suggests, these profiles suggest that K_D_ and Bmax cannot be accurately determined for the low affinity data (see red lines in the top row of Author response image 1). We wondered why GraphPad Prism did not detect any issues when fitting the low affinity data as it produces estimates for K_D_ and B_max_ without any cautionary prompts. This program is heavily used by the scientific community and it can detect parameter identifiable issues. We noticed that when plotting the same profiles but with a log-scale for the y-axis, a clear trough is visible even for the low affinity data (see red lines in the bottom row of Author response image 1). This likely explains why Prism does not flag the fit and highlights that interpreting these profiles in our case can be difficult and possibly subjective. On balance, given the target audience of our manuscript and the challenge of explaining profile likelihood and how they should be interpreted, we think that it may be easier to explain the issue with the more familiar CV.

**Author response image 1. sa2fig1:** Profiles for the data in main text Fig 1C for K_D_ (left) and B_max_ (right) when using a linear y-axis (top row) or log y-axis (bottom row). The y-axis is scaled such that the minima is at one, this facilitates comparing the profiles between high affinity and low affinity data sets.

Reviewer #2:1. Please further clarify and explain the extrapolation of the connection between B_max_ of the TRC-peptide binding curve and W6/32 binding curves based on Figure 1D.

We have modified the 1st Results section to explicitly discuss the quantitative relationship between B_max_ and W6/32 binding. This relationship is expected to be independent of the TCR affinity to pMHC because B_max_ is the amount of TCR binding when all pMHC are bound (which is independent of affinity) and W6/32 is a conformationally-sensitive antibody that does not depend on the precise peptide sequence. These two assumptions are borne out since we have used different affinity pMHCs to generate the standard curve.

1.1. Cosmetic – the color coding in 1B is a bit confusing because the same color palette denotes different concentrations of the same peptide here as the one denoting different peptides in 1D.

Thanks for pointing this out and we have now changed the colour scheme in Figure 1B

2. Please address the potential overfitting issue in the KPR model, and the parameter value spread in Figure S8.

We performed two independent fits of KPR; one fit to the plate data where we had 12 experiments with a total of 89 data points with 27 free parameters and one fit to the APC data where we had 17 experiments with a total of 126 data points with 37 free parameters. The ABC-SMC algorithm showed that N and all fitted kp values could be uniquely determined in both fits which are the two most important KP parameters. We now explicitly state this in the Results section on KPR.

3. Please comment on how the current model relates to antagonism, which is commonly considered a typical feature of TCR signaling

We have added a discussion paragraph to explain that the basic KP model cannot explain antagonism and have also taken the opportunity to explain that this model also does not explain an optimal pMHC affinity for T cell activation, which we and others have also reported. This is a limitation of the current model and we have highlighted that it would be interesting to extended the model to account for these in the future.

Reviewer #3:As came probably through already in the other review sections I feel enthusiastic about the work presented as it combines a massive volume of bench work-related data with relatively simple math (with the exception of the section dealing with KP) and it leads to testable clear hypotheses. The latter may be the tasks of others (or not), and hence I consider the manuscript close to being appropriate for publishing.Here are a few suggestions I consider useful for the reproducibility and readability by others.i. A detailed description of the SPR-protocol including primary data would render the reading more complete. The methods section is already fairly detailed, but it would be more than helpful to understand how the streptavidin and pMHC-bio run, as well as the TCR-runs and the final W6/32 run were computed against one another to arrive a the B_max_ of low affinity ligands. TCR-binding to pMHC (even if saturated) will result in a different SPR signal than antibody binding (given the different molecular masses). Since this methodology is central to the manuscript, I would argue in favor of introducing it more thoroughly in Figure 1A (and B, with representative raw data leading to corrected B_max_). This may also help to understand why the addition of W6/32 in both panels of figure 1B led to a much lower signal than the highest concentration of TCR.

We have now included a new paragraph in the 1st Results section and provided more details in the Methods.

ii. After lentiviral infection of primary T cells, how did the authors monitor the expression levels of the introduced TCRs? How did they verify the absence of mixed TCR dimers? Since TCR expression levels are clearly important in the overall scenario analyzed, I believe this is a fair question to ask.

We checked the expression of each batch of 1G4-transduced or electroporated T cells by staining with tetramers made with the NYE 9V pMHC to obtain the fraction of T cells expressing the transgenic TCR. As shown in Figure S2, due to the low level of A6 TCR expression this method led to a stark underestimation of the fraction of transgenic cells for the A6 TCR. We did not attempt to compare transgenic TCR molecules per cell between batches or in comparison to endogenous TCR. However, since we were able to reproduce our key findings with 2 TCRs expressed at very different levels, we believe that our findings (namely the value of *α*) are robust to changes in TCR expression levels.

We did not specifically exclude the formation of mixed TCR dimers by staining. Notably, untransduced T cells did not respond to any of the peptides we tested. While this does not formally rule out that mixed TCR dimers could contribute to responses to low affinity pMHCs, we think this unlikely given our robust correlation between K_D_ and potency and the fact that we could reproduce this phenotype in T cells derived from different human donors. Furthermore, the 1G4 TCR used for electroporation and the A6 TCR used for lentiviral transduction contained engineered cysteines to reduce potential mispairing (Cohen *et al.* Cancer Research. 2007.)

iii. I find it somewhat unusual (if not irritating) to find already published data in main figures. An in my opinion better approach is to cite the work and show exclusively the processed data (taken from cited work, in the lower panel of Figure 3A, B,C etc.).

We understand that this is unusual. We felt that on balance including it would allow us to engage a broader audience that may not have read these original manuscripts. By extracting the x,y coordinates of the data and re-plotting it, we feel that it provides a striking and intuitive result without the reader having to look up these papers themselves and provides confidence in our generated potency plots.

(iv) The Discussion section would benefit from a short paragraph reflecting on work on TCR-pMHC binding as measured within the confines of the immunological synapse. Reasons are that binding constants (if there is such a thing in synapses) differ in some cases substantially from what has been measured via SPR for reasons that are still being debated. Clearly, one major strength of this manuscript is that cellular behavior correlates with SPR measurements of TCR-pMHC binding. Nonetheless scenarios, especially those involving KP, may turn out differently in case serial TCR engagement by few antigens is supported.We have now included a new discussion paragraph explaining that there are occasional exceptions where the 3D binding parameters measured by SPR have not correlated with the T cell response and possible explanations for why this may be the case.

[Editors' note: further revisions were suggested prior to acceptance, as described below.]

They all appreciate the elaborated and thorough analyses presented in the paper, and also appreciate the additional work that had been done to address the previous comments of the reviewers. The manuscript has been improved, but there are some remaining issues that need to be addressed, as outlined below:1. One remaining concern mainly addresses the interpretation of the relationship between the values of B_max_ and the AB binding to pMHC in Figure 1D. The data suggest that this linear relationship could be extrapolated to very weak pMHC's but it does not need to be the case (see also comment reviewer 2). Although this is acknowledged in your statement (line 95-96), we recommend to discuss this in more detail. It seems that this assumption mainly drives the selection for the method to analyze the low affinities, with B_max_ constrained by this relationship. We would recommend to ackknowledge this in your discussion or provide additional evidence.

We apologise for this and believe that there is some confusion over what the TCR B_max_ (y-axis) and W6/32 binding (x-axis) are showing in Figure 1D and their relationship to the TCR/pMHC affinity. The B_max_ is the RU of TCR binding when all pMHC are bound, which is independent of TCR/pMHC affinity. It is the ability to experimentally estimate B_max_ that is practically limited by a low TCR/pMHC affinity. The W6/32 antibody binds all folded peptide-loaded human MHC class I molecules, independent of the specific peptide, and again, this is independent of TCR/pMHC affinity. In Figure 1D, we show that B_max_ and W6/32 correlate, which is expected and validates the use of W6/32 binding to estimate the B_max_.

The individual data points that make up Figure 1D include different pMHCs but also the same pMHC produced in different protein preparations and immobilised at different levels. This is why, for example, NYE 9V appears 6 times on this plot with a wide range of W6/32 binding and B_max_ values (larger values are obtained when more of this pMHC is immobilised on the surface). What the figures shows is that even when we have a large number of different pMHCs that bind their respective TCRs with different affinities, it is the level of pMHC immobilisation, not their affinity for a given TCR, that determines where along this line they reside. When we immobilised lower-affinity pMHCs, the W6/32 binding was always within the range used in this standard curve. Therefore, when determining their B_max_ we always *interpolated* within the standard curve but never extrapolated. This means that we always stayed within the linear part of the curve.

We’ve now added additional text and also labelled the amount of pMHC immobilised on the chip surface for NYE 9V to make this clear.

We hope that we understood the issue that was raised and that these changes improve the clarity.

2. There are some issues with regard to the presentation in Figure 1D that should be addressed.

There was an error in the labelling which is now fixed. This error has likely contributed to the confusion in understanding Figure 1D and we apologise for this and thank the reviewer again for bringing it to our attention.

Please also see the specific comments of the reviewers below.Reviewer #2 (Recommendations for the authors):I am still somewhat confused about the correspondence between the TCR B_max_ of binding to pMHCs in figure 1 B, but I think the importance of the results is compelling, and I am happy if this debate plays out in the literature rather than here. To summarize:1. It still looks (to me) that the linear relationship between Ab binding to pMHC and the B-max of the TCR binding is just an empirical finding, and although the data does suggest it might continue for very weak pMHC's I dont see why it would be guaranteed. On the same note, the authors say in their response, " W6/32 is a conformationally-sensitive antibody that does not depend on the precise peptide sequence" – which on its face contradicts the fact that the response to W6/32 is different to different peptides in figure 1 D.Please discuss.

We think the confusion is partly because of the error in the labelling of the legend in Figure 1D and we thank you for bringing this to our attention (it’s now fixed). The TCR B_max_ and W6/32 binding is not dependent on the specific peptide but rather how much correctly-folded pMHC has been immobilised on the surface. This is why NYE 9V appears 6 times in this plot with large variations of W6/32 binding and B_max_ (see response to point 1 above). Put differently, the reason W6/32 binding is different for different pMHCs is because there is a different amount of correctly folded pMHC on the chip surface even though in many cases we aimed to put a similar amount of pMHC on the chip surface (in some cases, such as for NYE 9V, we purposely put different amounts to check that W6/32 and B_max_ continued to be related). Whenever we used this curve to estimate B_max_ for lower-affinity pMHCs, the amount of W6/32 binding was always within the standard curve and therefore, we always stayed within the linear regime. We agree that had we immobilised very low levels of pMHCs (or very high levels) such that the W6/32 binding was outside of the linear regime, we would need to extrapolate and that could potentially introduce large errors (but we never did this).

2. I am still confused about the legend in Figure 1D. For instance, black dot is supposed to be Tax WT pMHC for A6 TCR – why are there four different black dots in the plot, with completely different B-max? Same question for NYE 9V. Finally filled orange dots are supposed to be Tax 1M. So what are the empty orange circles – they are not indicated in the legend, as far as I can see? The open purple circles appear in the plot but not in the legend. By contrast, open pink circle is indicated in the legend but does not seem to appear in the plot. Etc. If this is just mis-labeling it should be corrected and better explained in the text/caption.

As explained above, we performed experiments using the same pMHC immobilised at different levels on the chip surface and repeated the experiments when using independent protein preparations. This results in variations in the amount of correctly-folded pMHC that is on the chip surface that the TCR and the W6/32 antibody can recognise leading to variations in B_max_ and W6/32 binding RU. This confusion is a result of the incorrect legend labelling, which is now corrected and we apologise for this.

3. Figure 1C, left is supposed to show the binding curve for WT NY-ESO-1 pMHC but it is not shown in Figure 1B – why? Similarly, I could not find in the text what peptide is the Figure 1C, right for? In either case, it should be better explained in the text/caption.

Apologies for this and it is now corrected. The left panels (B,C) now show NYE 9V and right panels (B,C) show NYE 5F.